# Harnessing AlphaFold to reveal hERG channel conformational state secrets

**Khoa Ngo**[1,2]*, **Pei-Chi Yang**[1,2], **Vladimir Yarov-Yarovoy**[1,2,3], **Colleen E Clancy**[1,2,4]*, **Igor Vorobyov**[2,4]*

[1]Center for Precision Medicine and Data Science, University of California, Davis, Davis, United States; [2]Department of Physiology and Membrane Biology, University of California, Davis, Davis, United States; [3]Department of Anesthesiology and Pain Medicine, University of California, Davis, Davis, United States; [4]Department of Pharmacology, University of California, Davis, Davis, United States

*For correspondence:
khoango@ucdavis.edu (KN);
ceclancy@ucdavis.edu (CEC);
ivorobyov@ucdavis.edu (IV)

Competing interest: The authors declare that no competing interests exist.

## eLife Assessment

This **valuable** study uses AlphaFold2 to guide the structural modelling of different states of the human voltage-gated potassium channel $K_v11.1$, a key pharmacological drug target. Follow-up molecular dynamics and drug-docking simulations, combined with experimental characterization, offer **convincing** evidence supporting the models. The work shows potential for improving drug potency predictions in ion channel pharmacology.

**Abstract** To design safe, selective, and effective new therapies, there must be a deep understanding of the structure and function of the drug target. One of the most difficult problems to solve has been the resolution of discrete conformational states of transmembrane ion channel proteins. An example is $K_v11.1$ (hERG), comprising the primary cardiac repolarizing current, $I_{kr}$. hERG is a notorious drug anti-target against which all promising drugs are screened to determine potential for arrhythmia. Drug interactions with the hERG inactivated state are linked to elevated arrhythmia risk, and drugs may become trapped during channel closure. While prior studies have applied AlphaFold to predict alternative protein conformations, we show that the inclusion of carefully chosen structural templates can guide these predictions toward distinct functional states. This targeted modeling approach is validated through comparisons with experimental data, including proposed state-dependent structural features, drug interactions from molecular docking, and ion conduction properties from molecular dynamics simulations. Remarkably, AlphaFold not only predicts inactivation mechanisms of the hERG channel that prevent ion conduction but also uncovers novel molecular features explaining enhanced drug binding observed during inactivation, offering a deeper understanding of hERG channel function and pharmacology. Furthermore, leveraging AlphaFold-derived states enhances computational screening by significantly improving agreement with experimental drug affinities, an important advance for hERG as a key drug safety target where traditional single-state models miss critical state-dependent effects. By mapping protein residue interaction networks across closed, open, and inactivated states, we identified critical residues driving state transitions validated by prior mutagenesis studies. This innovative methodology sets a new benchmark for integrating deep learning-based protein structure prediction with experimental validation. It also offers a broadly applicable approach using AlphaFold to predict discrete protein conformations, reconcile disparate data, and uncover novel structure–function relationships, ultimately advancing drug safety screening and enabling the design of safer therapeutics.

## Introduction

Understanding the dynamic conformational changes of proteins is fundamental to elucidating their functions, interactions, and roles in biological processes. Many proteins, especially membrane proteins that constitute a significant portion of drug targets, exist in multiple functionally distinct states. Capturing these various conformations is crucial for predicting how proteins interact with ligands, designing drugs that selectively target-specific states, and uncovering the mechanisms that regulate these interactions. However, experimental techniques like cryo-electron microscopy (cryo-EM) often provide only static snapshots of proteins, typically capturing a single conformational state due to experimental constraints. Computational methods such as molecular dynamics (MD) simulations can sample alternative conformations but are limited by timescales and computational resources, often failing to observe meaningful conformational changes that result in functional effects. Enhanced sampling simulation techniques can help extend the timescales, but the biasing factors introduced to accelerate the simulations can sometimes push proteins into non-physiological conformations, potentially skewing the accuracy of the predictions and limiting their biological relevance.

Recent advances in deep learning have revolutionized protein structure prediction, with tools like AlphaFold2 (*Jumper et al., 2021a*) achieving remarkable success in predicting protein structures based on amino acid sequences. However, conventional applications of these AI-based methods often result in the prediction of a single, static conformation, akin to experimental snapshots. This raises a significant question: *Can we harness the capabilities of artificial intelligence to predict different physiologically relevant conformations of proteins, thereby capturing the dynamic spectrum of states essential for understanding protein function and drug interactions?*

To address this question, we employed and validated different strategies to guide AlphaFold2 to predict multiple physiologically relevant conformations, surpassing the usual single-state predictions. As a proof of concept, we applied this approach to the human voltage-gated potassium channel $K_V11.1$, encoded by the KCNH2 or human Ether-à-go-go-Related Gene (hERG) gene, a well-known drug anti-target in pharmacology and cardiology due to its role in drug-induced arrhythmias. hERG is a key player in cardiac electrophysiology, underpinning the rapid component of the delayed rectifier $K^+$ current ($I_{Kr}$) in cardiac myocytes (*Vandenberg et al., 2012*). This current plays a crucial role in the repolarization phase of the cardiac action potential (*Sanguinetti and Tristani-Firouzi, 2006*). Perturbation to hERG channel function, resulting from genetic anomalies or pharmacological interventions, can precipitate multiple arrhythmogenic disorders (*Sanguinetti and Tristani-Firouzi, 2006*).

The hERG channel is a homotetramer, with each subunit containing six membrane-spanning segments (S1–S6) (*Wang and MacKinnon, 2017*). The segments S5 and S6, along with the intervening loops and pore helix, form the channel pore domain (PD), crucial for potassium ion passage along the central pore, while segments S1–S4 form the voltage-sensing domain (VSD), responding to voltage changes across the cell membrane. Notably, the hERG channel also features specialized intracellular regions: the Per-Arnt-Sim (PAS) domain at the N-terminus and the cyclic nucleotide-binding domain at the C-terminus.

The distinctive pharmacological promiscuity of the hERG channel makes it prone to blockade by a diverse array of drugs, creating cardiac safety pharmacology risk in the drug discovery process. Blockade of the hERG channel by drugs can lead to QT interval prolongation known as acquired long QT syndrome (aLQTS) and escalate the risk of *torsades de pointes* (TdP), a potentially fatal arrhythmia (*Li and Ramos, 2017*). This issue has prompted the withdrawal of various drugs from the market and underscored the necessity of incorporating hERG safety evaluations in the drug development pipeline (*Ferri et al., 2013*; *Kocadal et al., 2019*; *Waldo et al., 1996*). The susceptibility for drug blockade is not uniform but varies depending on the channel conformational state, a phenomenon known as state-dependent drug block. Drugs may preferentially bind to and block the channel in specific states (open, closed, or inactivated), which can differentially affect cardiac repolarization and rhythm (*Perrin et al., 2008*; *Priest et al., 2008*) and thus confer different risks for aLQTS and TdP arrhythmias as shown in our previous study (*Yang et al., 2020*).

However, capturing the dynamic spectrum of hERG channel states poses a formidable challenge. While cryo-EM has offered invaluable insights into the putative open state of the channel (*Asai et al., 2021*; *Wang and MacKinnon, 2017*), a comprehensive view of the closed and inactivated states has remained elusive. Thus, even as we embark on a scientific era of explosive growth fueled by the convergence of protein structure insights, computational capabilities, and artificial intelligence based

modeling and synthetic data, the next frontier is marked by the need to reveal all relevant conformational states of proteins. The existing knowledge gaps constrain both predictive capabilities regarding drug–protein interactions and the creation of therapies through drug discovery to find specific and selective drugs, or in the case of hERG, to minimize their adverse interactions. For example, our recent study by Yang et al. introduced a multiscale model framework to forecast drug-induced cardiotoxicity at cellular and tissue levels, utilizing atomistic simulations of drug interactions with the hERG channel (*Yang et al., 2020*). However, the absence of hERG structural models in the inactivated and closed states limited the predictive potential of atomistic scale simulations of state-specific drug binding.

The emergence of AlphaFold2, a protein structure prediction tool driven by machine learning, has brought a paradigm shift in structural biology (*Jumper et al., 2021a*). AlphaFold2 represents a significant advance over previous methods by using deep learning to predict the three-dimensional structures of proteins (*Jumper et al., 2021a*). AlphaFold2 primarily requires a protein's amino acid sequence as input, but it also leverages other critical data sources. In addition to the sequence, it incorporates multiple sequence alignments (MSAs) of related proteins from different species, available structural templates, and information on homologous proteins (*Jumper et al., 2021a*). While the primary sequence encodes the 3D structure, AlphaFold2 harnesses evolutionary conservation from MSAs to reveal structural insights that extend beyond what a single protein sequence can provide. These additional inputs help the model to identify evolutionary and structural constraints that are crucial for accurate predictions. The output of AlphaFold2 is a predicted 3D structure of the protein that includes inter-residue distance predictions, whereby the model predicts the distances between every pair of amino acid residues in the protein. Predictions about the angles between bonds that connect amino acid residues are also generated as angle predictions that are crucial for determining the precise shape of the protein fold.

AlphaFold2's limitation, in its default configuration, is that it generates only a single-state structure (*Lane, 2023*), which for the hERG channel corresponds to the open state. In this study, we introduced an easily replicable and generally applicable approach to guide AlphaFold2 in predicting multiple, physiologically relevant conformations of proteins. By employing multiple structural templates and refining input parameters, we enhanced the predictive capabilities of AlphaFold2, enabling it to generate highly relevant and physically plausible protein conformations beyond the default single-state prediction. We conducted drug docking simulations to predict how specific drugs interact with the hERG channel in different conformational states and performed MD simulations to assess ion conduction across these states. Throughout the process, we validated our predictions by comparing them with experimental data, ensuring that both the drug interactions and ion conductive properties aligned with observed experimental outcomes. This method opens new possibilities for in silico studies of protein dynamics, drug design, and safety assessments, allowing researchers to explore the full range of conformational states that proteins may adopt.

## Results
### Generating hERG channel conformational states

It is well known that the hERG channel resides in discrete functional states, minimally comprising closed, open, and inactivated states, which interconnect as a function of time and membrane voltage (*Vandenberg et al., 2012*). In the open state, the channel conducts $K^+$ ions through a central pore. In contrast, the closed and inactivated states are non-conductive due to either a constricted pore at the intracellular gate (closed state) or a distorted selectivity filter (SF; inactivated state) (*Vandenberg et al., 2012*). So far, published experimental cryo-EM structures resolved the channel in an open state (*Asai et al., 2021*; *Wang and MacKinnon, 2017*). Starting with the experimental structure, computational studies explored hERG inactivation by simulating how different membrane voltages can change the SF and thus affect ion conduction through the channel (*Li et al., 2021b*; *Miranda et al., 2020*; *Yang et al., 2020*). These studies are essential but have some limitations, as the high voltages applied can force the channel into unnatural conformations, and the simulations are not long enough to allow observation of state transitions (*Shi et al., 2020*). To overcome these limitations, we adopted different modeling strategies to guide AlphaFold2 in producing diverse conformations relevant to specific functional states of the hERG channel.

The first modeling strategy involves using a structural fragment from an experimental structure of a homologous protein that exhibits the desired characteristics of the target state we aim to model in our channel. This fragment serves as a structural template, and AlphaFold2 is used to rebuild the rest of the channel while adhering to the constraints of the template. For example, to model the closed state of the hERG channel, it is known that channel closure requires the voltage sensor in the voltage-sensing domain to be in a deactivated conformation. To achieve this, we used the deactivated voltage-sensing domain from the closed-state rat EAG channel cryo-EM structure (PDB 8EP1, residues H208–H343) (*Mandala and MacKinnon, 2022*) and combined it with the SF from the open-state hERG cryo-EM structure (PDB 5VA2, residues I607–T634) (*Wang and MacKinnon, 2017*). This hybrid structure was used as the template for AlphaFold2 predictions, as illustrated in *Figure 1a*. Using these discrete structural fragments, AlphaFold2 was then applied to generate 100 models, specifically configured to encourage diverse prediction outcomes for further analysis.

For modeling the open hERG channel state, we utilized the existing cryo-EM structure of hERG (PDB 5VA2) (*Wang and MacKinnon, 2017*) and rebuilt the missing extracellular loops using Rosetta (*Fleishman et al., 2011*) with the results shown in *Figure 1b*. This reconstructed model served as a basis for MD and drug docking simulations.

The second modeling strategy addresses situations where structural information about state transitions is either limited or inconsistent. In this approach, we erase regions expected to undergo changes during state transition from an existing structure and use AlphaFold2 to sample potential conformations for these regions. This allows AlphaFold2 to identify possible substates, which are then grouped into clusters of structurally similar models for further analysis. For example, during hERG inactivation, the SF shifts from an open, conductive to a distorted, non-conductive conformation, as shown by numerous studies on hERG and other K$^+$ channels (*Butler et al., 2019*; *Cuello et al., 2010*; *Fan et al., 1999*; *Li et al., 2021b*; *Pettini et al., 2023*; *Schönherr and Heinemann, 1996*; *Tan et al., 2022*; *Wu et al., 2025*). Moreover, there are a number of studies that do not uniformly suggest a single discrete structure of the inactivated-state SF but propose several alternative conformations (*Lau et al., 2024*; *Li et al., 2021b*).

To model the inactivated state of hERG, first we configured AlphaFold2 to introduce more uncertainty into the sampling process. As illustrated in *Figure 1c*, starting with the open-state cryo-EM structure (PDB 5VA2) (*Wang and MacKinnon, 2017*), we removed everything except for the cytosolic domain (S660–R863), then let AlphaFold2 reconstruct the transmembrane domain. In half of the resulting predictions, including the top-ranked model by prediction confidence, the SF showed a distinct lateral flip of the backbone carbonyl oxygens at residue V625 compared to the open-state structure. This flip created a potential barrier that could prevent K$^+$ ions from crossing between the S3 and S2 ion-binding sites. To further investigate this conformation, we extracted the predicted SF region (residues Y607–T634) and merged them with the activated VSDs (W398–V549) and the cytosolic domain (S660–R863) from the open-state hERG structure (PDB 5VA2) (*Wang and MacKinnon, 2017*) to create a new structural template. Then, we generated 100 more models for further analysis.

## Clustering of AlphaFold2-generated hERG models reveals predominant substates

A key distinction of our approach is that, rather than relying solely on single-model predictions, we generated a diverse population of models to better explore the conformational landscape. By clustering these models, we identified predominant substates, represented as clusters of structurally similar models. To determine which of these substates are likely to be physiologically relevant, we quantitatively assessed the structural reliability within each cluster using the predicted Local Distance Difference Test (pLDDT). Higher pLDDT scores indicate more reliable and accurate structural predictions (*Jumper et al., 2021a*). This clustering approach helps to capture a range of conformations that might represent stable states of the protein.

For each predicted conformational state of hERG, we clustered 100 predicted structural models based by their degree of similarity, quantified by the root-mean-square deviation (RMSD), as shown in *Figure 2*.

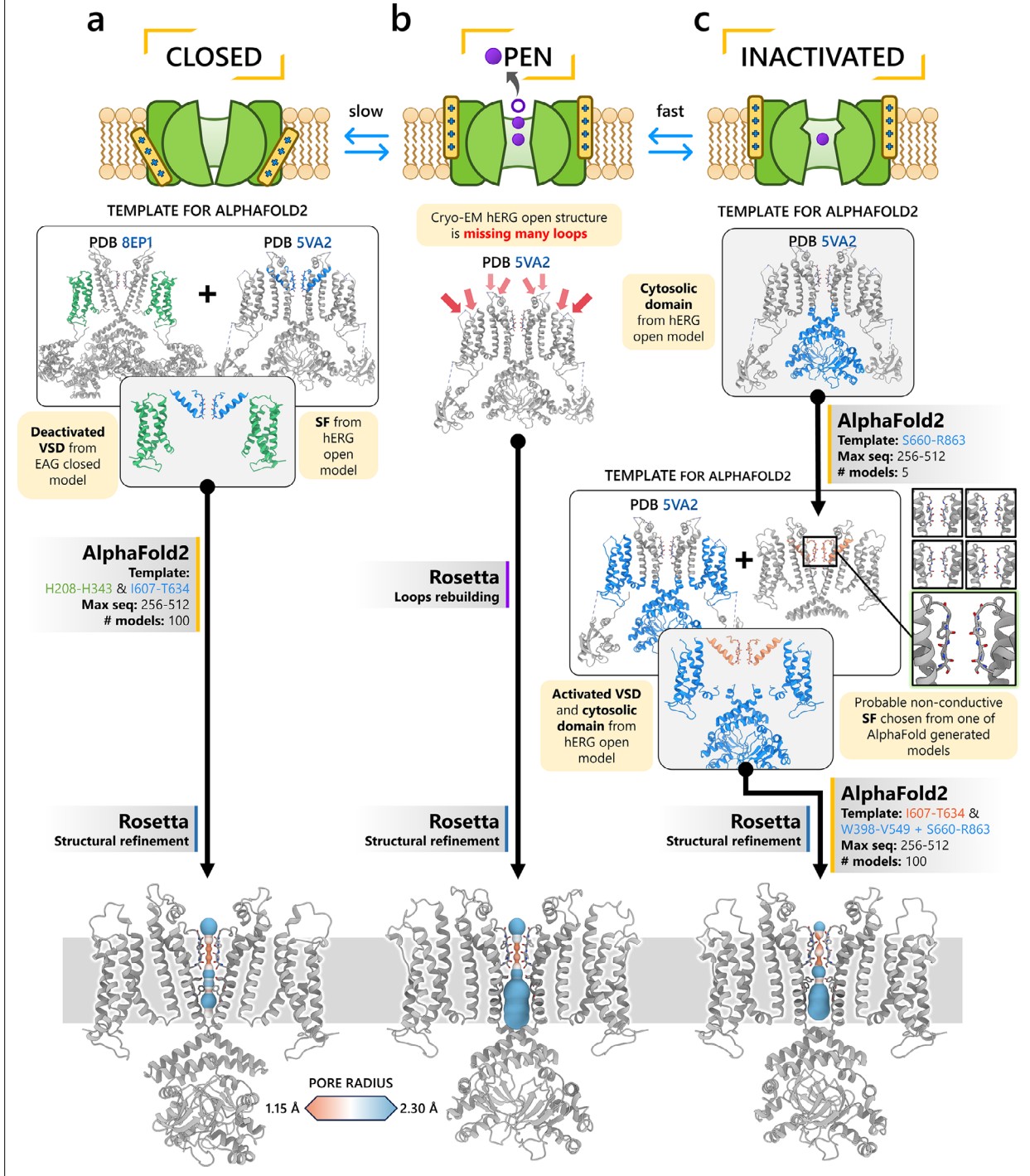

**Figure 1.** Generation of hERG channel models in the closed (**a**), open (**b**), and inactivated (**c**) states. The lower limit of the pore radius color profile (1.15 Å) indicates the minimum radius to accommodate a water molecule, and the upper limit (2.30 Å) indicates sufficient space to fit two water molecules side-by-side. 'Max seq' is a setting in ColabFold that denotes the maximum number of cluster centers and extra sequences that the multiple sequence alignment (MSA) used for AlphaFold2 will be subsampled to. '# models' indicates the number of models predicted using the provided structural templates.

The online version of this article includes the following figure supplement(s) for figure 1:

**Figure supplement 1.** An example illustrating the extension of our strategy to model alternative ion channel states using AlphaFold.

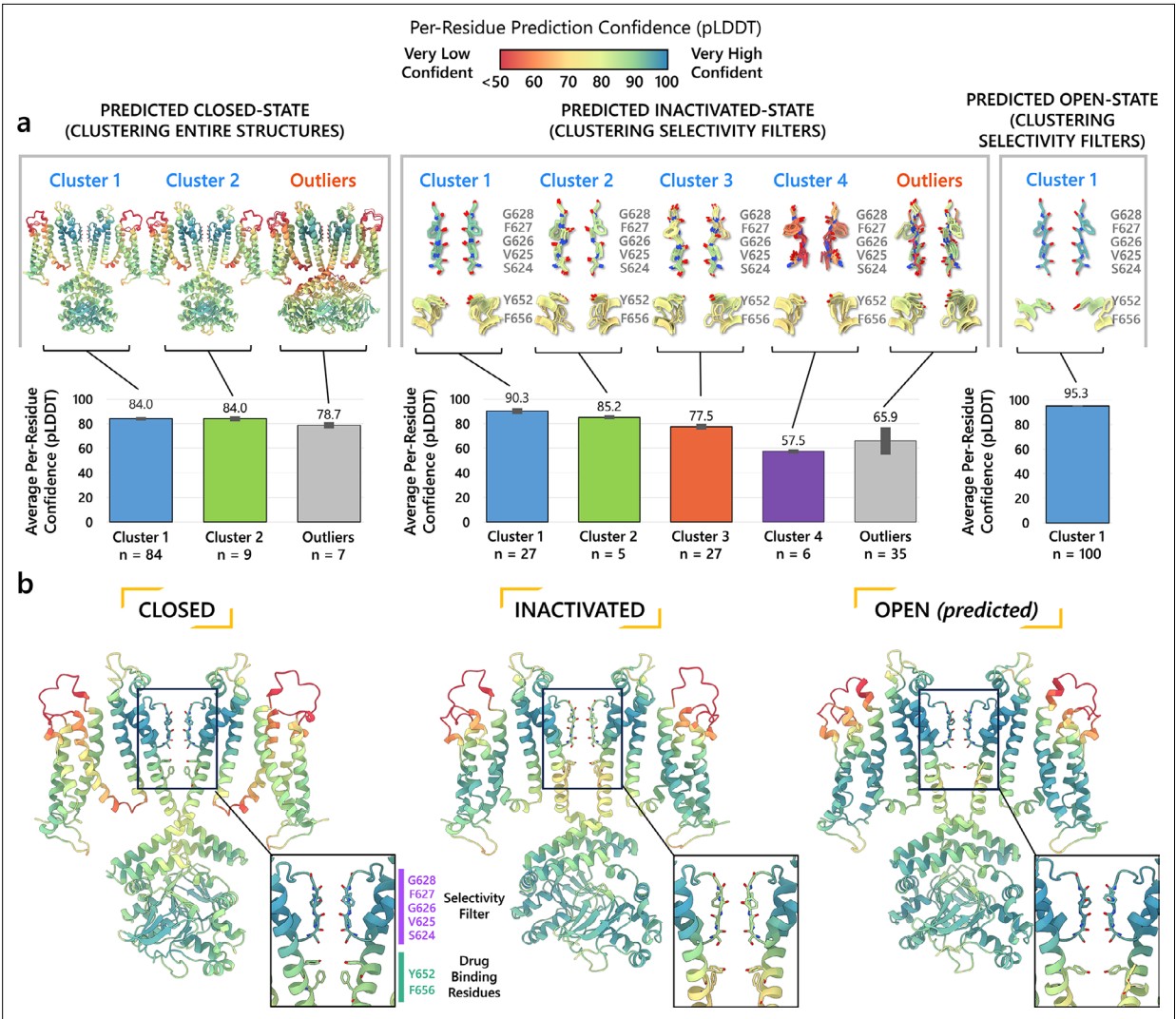

**Figure 2.** Clustering of AlphaFold2-predicted hERG channel models. (**a**) Clusters created from 100 models predicted for each state. Each structure visualized is colored according to the per-residue confidence metric (predicted Local Distance Difference Test, pLDDT). The closed-state models are clustered based on the backbone $C_\alpha$ RMSD of the entire protein models. The inactivated and open-state models are clustered based on the all-atom RMSD of the selectivity filter (residues S624–G628). To represent each cluster, the top 5 models ranked by an average pLDDT are shown. The bar graphs display the mean pLDDT values for the clustered segments across all models within each cluster, with the standard deviations shown as error bars. Clusters containing less than three models are categorized as outliers. (**b**) The models chosen for subsequent analysis colored by per-residue pLDDT values.

The online version of this article includes the following figure supplement(s) for figure 2:

**Figure supplement 1.** Protein backbone dihedral angle distributions of AlphaFold predicted models during inactivated-state-sampling and reference structures across key residues involved in ion selectivity and drug binding.

**Figure supplement 2.** Comparison of the SF in hERG closed- (**a, d**), open- (**b, e**), and inactivated-state (**c, f**) models.

## Closed-state clusters

The analysis of closed-state clusters showed only minor differences in RMSD and pLDDT values between them. When comparing the top-ranked models from each cluster, the all-atom RMSD between Cluster 1 (the cluster with the highest confidence) and Cluster 2 was just 0.36 Å, while the RMSD between Cluster 1 and the outlier cluster was 0.95 Å (outlier clusters are those with fewer than three members). This indicates small structural differences among the models. Aside from the outlier cluster, the average pLDDT scores for Clusters 1 and 2 were also very similar. The low RMSD values suggest that the predictions are converging on a similar overall conformation, with the minor

differences likely due to slight variations in the positioning of the intracellular loop regions. As a result, the top-ranked model from Cluster 1 (*Figure 2b*) was selected for further simulations.

## Inactivated-state clusters

As inactivation is known to affect the SF, we grouped the models by focusing exclusively on similarity of the SF (S624–G628) conformations. We ranked the clusters according to the average pLDDT of these specific residues. This method led to the identification of four main clusters and one outlier. Cluster 1, which has the highest confidence score (pLDDT = 90.3 ± 2.3, *n* = 27), contains models in which most SF carbonyl oxygens point inward, that is, toward the central axis, resembling an open-state SF conformation. In contrast, Cluster 2 (pLDDT = 85.2 ± 1.6, *n* = 5) is distinguished by the outward flipping of the V625 carbonyl and a noticeable pore narrowing between the G626 carbonyls. Cluster 3 (pLDDT = 77.5 ± 2.5, *n* = 27) is characterized by reorientation of the G626 carbonyls and, in rare cases, those of F627 residues. Cluster 4 (pLDDT = 57.5 ± 1.7, *n* = 6) exhibits a mixed conformation that combines features of both Clusters 2 and 3, along with occasional rearrangement of S624 residues, although this rearrangement introduces steric clashes with neighboring side chains. In most models, particularly those in Clusters 2 and 3, the S6 helix undergoes varying degrees of rotation, leading to repositioning of the pore-lining drug-binding residues Y652 and F656, whose side chains extend further into the central cavity. The remaining models display SF conformations with varying combinations of features from previous clusters, but due to subunit-to-subunit variability, these were grouped as outliers.

Interestingly, the inactivated-state SF conformations predicted by AlphaFold coincide with proposed hERG C-type inactivation mechanisms as highlighted in other experimental and computational studies (*Lau et al., 2024*; *Li et al., 2021b*). Specifically, the flipping of V625 and the constriction at G626 carbonyls in Cluster 2 was previously reported in a recent study (*Lau et al., 2024*). Moreover, Li et al.'s computational work revealed an asymmetric SF conformation, where two opposing subunits exhibited similar V625 flipping and G626 narrowing characteristics, while the other two subunits displayed the G626 and F627 carbonyl reorientation characteristic of our Cluster 3 (*Li et al., 2021b*). Remarkably, AlphaFold2 was able to independently predict these conformations, despite the fact that they were not part of its training dataset, which had a cutoff year of 2021 (*Jumper et al., 2021a*) and did not include simulated models.

To assess conformational variability, we examined backbone dihedral angles (phi φ and psi $\psi$) at key residues in the SF (S624–G628) and drug-binding region on the pore-lining S6 segment (Y652 and F656), of all 100 models sampled here as shown in *Figure 2—figure supplement 1*. By overlaying the φ and $\psi$ dihedral angles from different models, including the open state (PDB 5VA2-based), the closed state, and representative models from AlphaFold inactivated-state-sampling Clusters 2 and 3, we found that these conformations consistently fall within or near high-probability regions of the dihedral angle distributions. This indicates that these structural states are well represented within the ensemble of conformations sampled by AlphaFold within the scope of this study, particularly at functionally critical positions.

Since the SF conformation in Cluster 2 has been observed experimentally (*Lau et al., 2024*), but its overall pore architecture differs from our models, we selected the highest confidence model from this cluster as the representative model for initial structural comparisons and analyses. This model consistently shows flipping of SF residue V625 carbonyls, pinching (decreased distance) between SF residue G626 carbonyls, and rearrangement of drug-binding residues across all subunits. To broaden our assessment of potential inactivated structural models, we also conducted molecular simulations with the top model from Cluster 3 to evaluate its conformational behavior and functional relevance.

## Open-state clusters

As a control, we combined parts of the presumed open-state cryo-EM hERG structure (PDB 5VA2) (*Wang and MacKinnon, 2017*), specifically the conductive SF, the activated VSDs, and the cytosolic domain, as the structural template for AlphaFold2 to test whether it would predict changes to the pore region similar to those observed in other states' predicted clusters. Post-prediction, all 100 generated models are nearly identical, converging almost uniformly into a single cluster. The highest scoring model closely mirrors the experimental open-state cryo-EM structure (*Wang and MacKinnon, 2017*), with a virtually identical pore region. Given the minimal differences between the open-state models

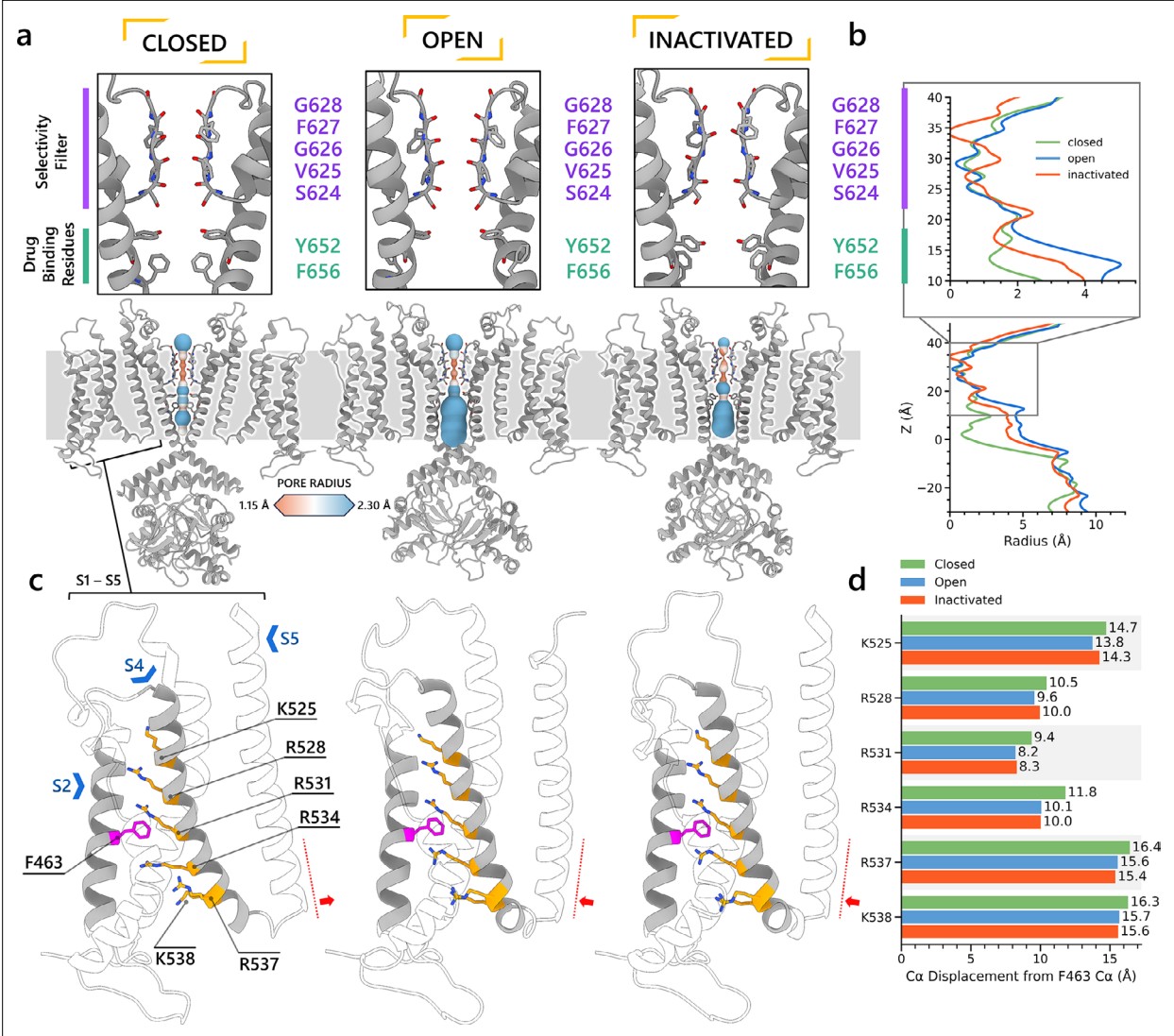

**Figure 3.** Structural comparison of different hERG channel state models. (**a**) Visual comparison of the closed-, open-, and inactivated-state models. (**b**) Pore radius for the selectivity filter (SF) and drug-binding region (upper) and for the entire pore (lower). (**c**) Comparison of the voltage sensing domain (VSD) conformation in each model, showcasing the positively charged Arg and Lys gating-charge residues (yellow), located on the S4 helix, and the gating charge transfer center residue, F463 (magenta), on the S2 helix. (**d**) Distances between the $C_\alpha$ atom of residue F463 to the $C_\alpha$ atom of each of the gating-charge residues.

The online version of this article includes the following figure supplement(s) for figure 3:

**Figure supplement 1.** Distance-based contact maps comparing intra- and intersubunit contacts between each model.

**Figure supplement 2.** Comparison of the S6 helix conformation for the hERG closed- (**a**), open- (**b**), and inactivated-state (**c**) models.

with Rosetta-rebuilt and AlphaFold-predicted loops, we would not expect any significant impact on our results had either been used. For consistency with prior studies and to facilitate direct comparison, we selected the experimental cryo-EM structure (PDB 5VA2) with loops rebuilt by Rosetta to represent the open state, as this structure and approach have been widely used as an open-state reference in our previous hERG channel studies (*Miranda et al., 2020*; *Yang et al., 2020*). As such, no models from this prediction were considered for further testing.

# Comparison of hERG channel state models reveals structural differences in the SF and channel pore

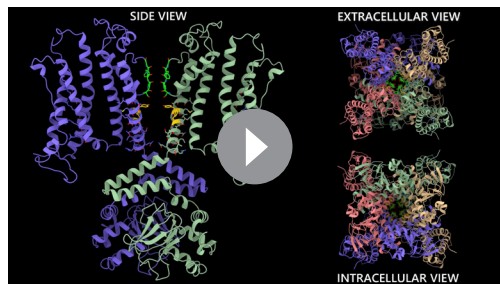

After further structural refinement in Rosetta (*Fleishman et al., 2011*; *Leman et al., 2020*) to resolve steric clashes, the resulting models are compared in *Figure 3*. In *Figure 3a, b*, the closed-state model displayed the most constricted channel pore, followed by the inactivated state and then the open-state model. In the pore-lining S6 helix, the canonical drug-binding residue Y652 (*Vandenberg et al., 2012*) retains a relatively consistent position with minor variation across all channel state models. The rotation and shift of the S6 helix in the inactivated and closed states affect the position of another canonical drug-binding residue, F656 (*Vandenberg et al., 2012*). The adjustment caused the F656 side chain to extend more into the hERG inner cavity in both the closed and inactivated states, compared to the open state.

**Video 1.** Animation depicting the hERG channel transitioning through various states, beginning in the open state and ending in the closed state using the structural models developed in this study.
https://elifesciences.org/articles/104901/figures#video1

## Selectivity filters

Shown in *Figure 2—figure supplement 2a, b*, the SFs of the open- and closed-state models display similar conformations, with carbonyl oxygens along the ion path all oriented toward the central axis as in other K⁺ channel structures, for example, KcsA and $K_V$1.2, enabling efficient knock-on K⁺ conduction (*Doyle et al., 1998*; *Long et al., 2005*). In contrast, the inactivated-state model SF is distinct, marked by the lateral rotation of the V625 backbone carbonyls away from the central axis (*Figure 2—figure supplement 2c*), thereby creating a potential barrier preventing ion crossing. Additionally, we noted a constriction between the G626 backbone carbonyls and a repositioning of the S624 side chain hydroxyl oxygens. In the model representing the inactivated state, the carbonyl oxygens of G628 and F627 exhibit an upward shift relative to their positions in the open-state model. *Figure 2—figure supplement 2d-f* presents the SF of all three models from an extracellular perspective. In both the closed- and inactivated-state models, the F627 side chain undergoes a clockwise rotation when contrasted with its orientation in the open state. This rotational behavior aligns with findings from a prior simulation study (*Miranda et al., 2020*) where it was noted in a metastable non-conductive state. The loop that links the upper SF to the S6 helix rotates anti-clockwise relative to its position in the open-state model, consequently narrowing the upper part of the SF.

## VSDs

For the VSD, we measured the distances between the backbone $C_\alpha$ atoms of the gating charge residues (K525, R528, R531, R534, R537, and K538) on the S4 helix and the gating charge transfer center residue F463 on the S2 helix, as shown in *Figure 3c*. Although we observed an increased distance between the gating charge residues and the charge transfer center residue in the closed state, this separation was not due to a straight downward movement of the S4 helix. Instead, the closed-state model S4 exhibited a minor kink around residue R531 and lateral movement toward the channel center, impacting the S4–S5 linker and consequently nudging the S5 helix inward, effectively narrowing the pore. The predicted closed-state model exhibits lower confidence levels for the S4 helix and S4–S5 linker residues (pLDDT ≤75) when compared to their counterparts in models of other states, necessitating caution in interpreting the physiological implications of this observation. Conversely, the pore region, which demonstrates closure, is characterized by a higher prediction confidence (pLDDT ≥75), suggesting a more robust and reliable structural representation. *Video 1* shows an animation of state changes of the hERG channel models.

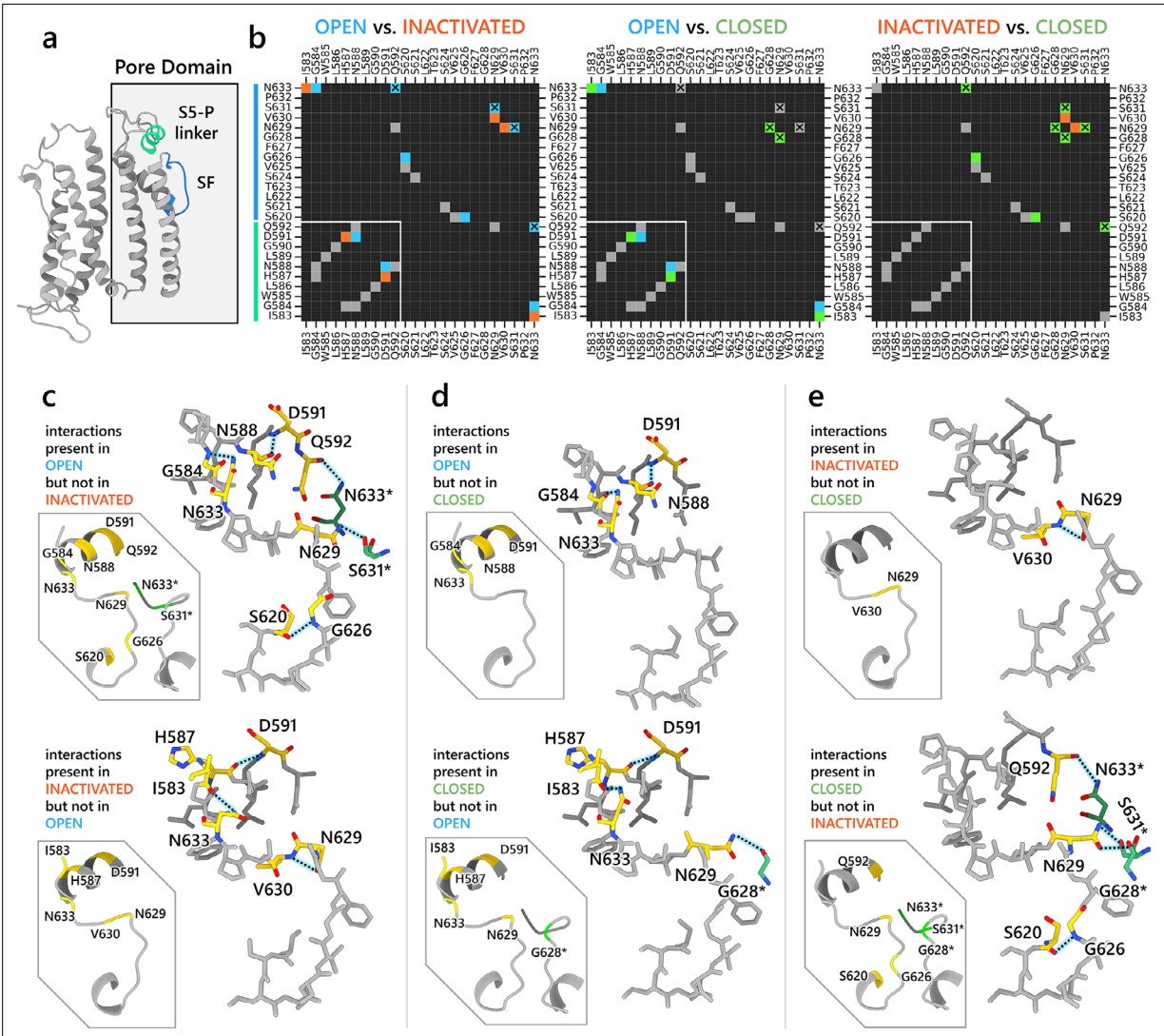

**Figure 4.** Interaction network analysis showcasing residue–residue interactions in the S5-P linker (residues I583–Q592) and region surrounding the selectivity filter (SF) (residues S620–N633). (**a**) An image of a hERG channel subunit with the analyzed S5-P linker and SF regions colored in light green and light blue, respectively. (**b**) Heatmaps showing intrasubunit and intersubunit (marked by X) interactions between each residue in the analyzed regions. The interactions analyzed are hydrogen bonding, π stacking, cation–π, and salt bridges. Black cells indicate no interactions. Gray cells indicate an interaction is present in both states. Blue, orange, and green colored cells indicate the interaction is present only in the open, inactivated, or closed state, respectively, but not in the other state being compared in the map. White lines are added to separate S5-P linker residues from the SF region residues. (**c–e**) Visualization of the interactions being present in one state but not the other. Gold-colored residues are involved in the interactions. Green-colored residues, named with an asterisk at the end, are from an adjacent subunit but are interacting with gold-colored residues. Dashed lines represent hydrogen bonds.

## Interaction networks

We aimed to further investigate the molecular interactions that contributed to channel inactivation through modulation of the SF conformation. In *Figure 4*, we analyzed residue-residue interactions of extracellular S5-P linker (I583–Q592) along with SF and surrounding SF residues (S620–N633, *Figure 4a*) through heatmaps detailing hydrogen bonding, π stacking, cation–π, and salt bridge formation similarities and differences (*Table 1* shows detection criteria; distance-based contact maps (*Noel et al., 2012*) for all residues are shown in *Figure 3—figure supplement 1*). Distinct interaction patterns between open- and inactivated-state models were observed in these regions (*Figure 4b, c*). In the open-state model, N633 atop the SF forms hydrogen bonds with S5-P linker G584 from the same and Q592 from an adjacent subunit, while N629 forms hydrogen bonds with an adjacent-subunit S631. Additionally, SF G626 forms an intra-subunit hydrogen bond with S620 behind the SF. Within

**Table 1.** Criteria for different types of non-bonded interactions used in analyses.

| Types of interaction | Detection criteria | Filtering criteria |
|---|---|---|
| Hydrogen bonds | Distance between the hydrogen bond donor (D) and acceptor (A) should be less than 3.8 Å. The angle D-H...A should be above 110°. | Hydrogen bonds between atoms that already form a salt bridge are excluded. A hydrogen bond donor can participate in only one hydrogen bond, while acceptor atoms can form multiple hydrogen bonds. |
| Salt bridges | Geometric centers of oppositely charged groups that come within 4.5 Å. | |
| Cation–π | Pairing of a positive charge and an aromatic ring if the distance between the charge center and the aromatic ring center is less than 6 Å. | |
| π-Stacking | Geometric centers of two aromatic rings within 5.5 Å. The angle between the rings should deviate no more than 30° from the optimal angle (90° for T-stacking, 180° for P-stacking). When projecting each ring center onto the opposite ring plane, the distance between the other ring center and the projected point (offset) should be less than 2 Å. | |

the S5-P linker, N588 and D591 also display hydrogen bonding. However, these stabilizing interactions in the open-state model SF region are absent in the inactivated-state model, where only intra-subunit hydrogen bonds between I583 (S5-P linker) and N633 (SF) occur, along with V630 hydrogen bonding with the same-subunit N629 atop the SF. To corroborate our findings, mutations involving the residues discussed above have been shown to impact hERG inactivation as evidenced in numerous clinical and experimental studies (*Butler et al., 2018*; *Clarke et al., 2006*; *Cordeiro et al., 2005*; *Dun et al., 1999*; *Fan et al., 1999*; *Ficker et al., 1998*; *Miranda et al., 2020*; *Nakajima et al., 1998*; *Satler et al., 1998*) (see *Table 2* for more details).

The open- and closed-state models show fewer differences in their SF hydrogen bond networks compared to those between the open and inactivated states (*Figure 4b, d*). In the open-state model, D591 from the S5-P linker forms an intra-subunit hydrogen bond with N588, and G584 hydrogen-bonds with N633 at the top of the SF. These interactions are absent in the closed-state model, where H587 (instead of N588) hydrogen-bonds with D591 within the S5-P linker, and I583 (replacing G584) interacts with N633 at the SF top. Additionally, G628 from an adjacent subunit forms a hydrogen bond

**Table 2.** Mutations known to affect hERG channel inactivation.

| Mutation(s) | Reported impact on hERG channel | References |
|---|---|---|
| N629D, N629S, N633S | Disrupt inactivation and K⁺ selectivity | *Lees-Miller et al., 2000b*; *Satler et al., 1998* |
| H587P/K | Disrupts C-type inactivation and K⁺ selectivity | *Dun et al., 1999* |
| S631A | Causes positive shift in half-inactivation voltage | *Butler et al., 2018*; *Zou et al., 1998* |
| S631C | Speeds up fast inactivation | *Fan et al., 1999* |
| N588K/E, Q592K | Modulate rapid inactivation | *Clarke et al., 2006*; *Cordeiro et al., 2005* |
| D591R/Q592R | Inhibit inactivation | *Clarke et al., 2006* |
| G584S | Leads to inactivation gating defects | *Zhao et al., 2009* |
| S620T | Abolishes hERG inactivation | *Ficker et al., 1998* |
| V630L | Causes negative shift in steady-state inactivation | *Nakajima et al., 1998* |
| N588C, I583C | High and intermediate impact on hERG inactivation | *Liu et al., 2002* |

with N629 atop the SF. Analyzing the differences between the inactivated- and closed-state model (*Figure 4e*), the inactivated-state model uniquely features an intra-subunit V630–N629 hydrogen bond, whereas the closed-state model exhibits intersubunit hydrogen bonds between N633 and Q592, and between S631/G628 and N629. Furthermore, in the closed-state model, S620 forms an intra-subunit hydrogen bond with G626, stabilizing the SF conformation.

## S6 pore-lining helix

In *Figure 3—figure supplement 2*, we compared the S6 helix orientation across various models. The closed-state model features a mostly straight S6 helix. On the contrary, both the open- and inactivated-state models exhibit a pronounced kink around I655, as identified in a prior study (*Thouta et al., 2014*), which facilitates pore opening and distinguishes the inactivated state from the closed-state model. Notably, a slight rotation differentiates the S6 helix in the open- and inactivated-state models, altering the conformation of drug-binding residues Y652 and to a greater extent, F656. The interaction network analysis results from *Figure 4b, c* suggest that alterations in the hydrogen bond network around the SF region, during the transition from open to inactivated state, might pull on the S6 helix and influence its orientation (*Figure 3—figure supplement 2b, c*) – a subtle yet potentially impactful change for drug binding. In agreement with our observations, a study by Helliwell et al. also suggested that a slight clockwise rotation of the S6 helix in the hERG open-state cryo-EM structure (*Wang and MacKinnon, 2017*) could align the S6 aromatic side chains, particularly F656, into a configuration enabling interactions with inactivation-dependent blockers that more accurately reflects experimental data (*Helliwell et al., 2018*).

## MD simulations show K$^+$ ion conduction in the open-state model but not in the inactivated state

We performed all-atom MD simulations on two hERG channel models described above, one in the open state and the other in the predicted inactivated state, to evaluate their ion conduction capabilities. Unlike the closed-state model, both open- and inactivated-state models should allow ions and water to enter and traverse the channel pore reaching the SF region. However, only the open-state model is expected to facilitate ion conduction through its SF.

### Ion conductivity

To investigate ion conduction in the SF, we considered two conditions, as shown in *Figure 5—figure supplement 1a* one in which the SF initially contained only K$^+$ ions and another in which both ions and water were present to test previously proposed direct (or Coulombic) and water-mediated K$^+$ conduction knock-on mechanisms (*Lam and de Groot, 2023*; *Roux, 2017*) as in a previous study (*Miranda et al., 2020*). In the direct knock-on (ions-only) scenario, we manually positioned K$^+$ atoms in the putative K$^+$-binding sites of S0, S2, S3, and S4 within the SF. For the water-mediated knock-on (the alternating ions and water molecules) scenario, K$^+$ ions were placed in the S1, S3, and S$_{cav}$ positions, while water molecules were inserted into the S0, S2, and S4 positions. These models were incorporated into phospholipid bilayers consisting of *1-palmitoyl-2-oleoylphosphatidylcholine* (POPC) molecules and hydrated by 0.30 M KCl, as depicted in *Figure 5—figure supplement 1b*. Subsequently, we conducted MD simulations for each case under three membrane voltage conditions: 0, 500, and 750 mV, each lasting 1 µs. This resulted in a total of six MD simulations for each model.

In all instances where a non-zero membrane voltage was applied after equilibration, we observed K$^+$ conduction for the open-state model (*Figure 5a, c* and *Figure 5—figure supplement 2a, c*), whereas such conduction was not observed for the inactivated-state model (*Figure 5b, d* and *Figure 5—figure supplement 2b, d*). For ions-only initial SF arrangement, we observed that all K$^+$ ions initially located in the SF went across during 1 µs MD runs under applied 750 and 500 mV membrane voltages (*Figure 5a* and *Figure 5—figure supplement 2*), whereas for the alternating water-ion initial SF configuration, we observed conduction of SF ions as well as additional K$^+$ ions moving all the way across the channel pore (*Figure 5c* and *Figure 5—figure supplement 2c*). In both cases, we saw a combination of direct and water-mediated knock-on mechanisms, as in our previous hERG channel MD simulations (*Miranda et al., 2020*; *Yang et al., 2020*). Control MD simulations conducted under zero voltage conditions revealed a single K$^+$ SF conduction event for the open-state model

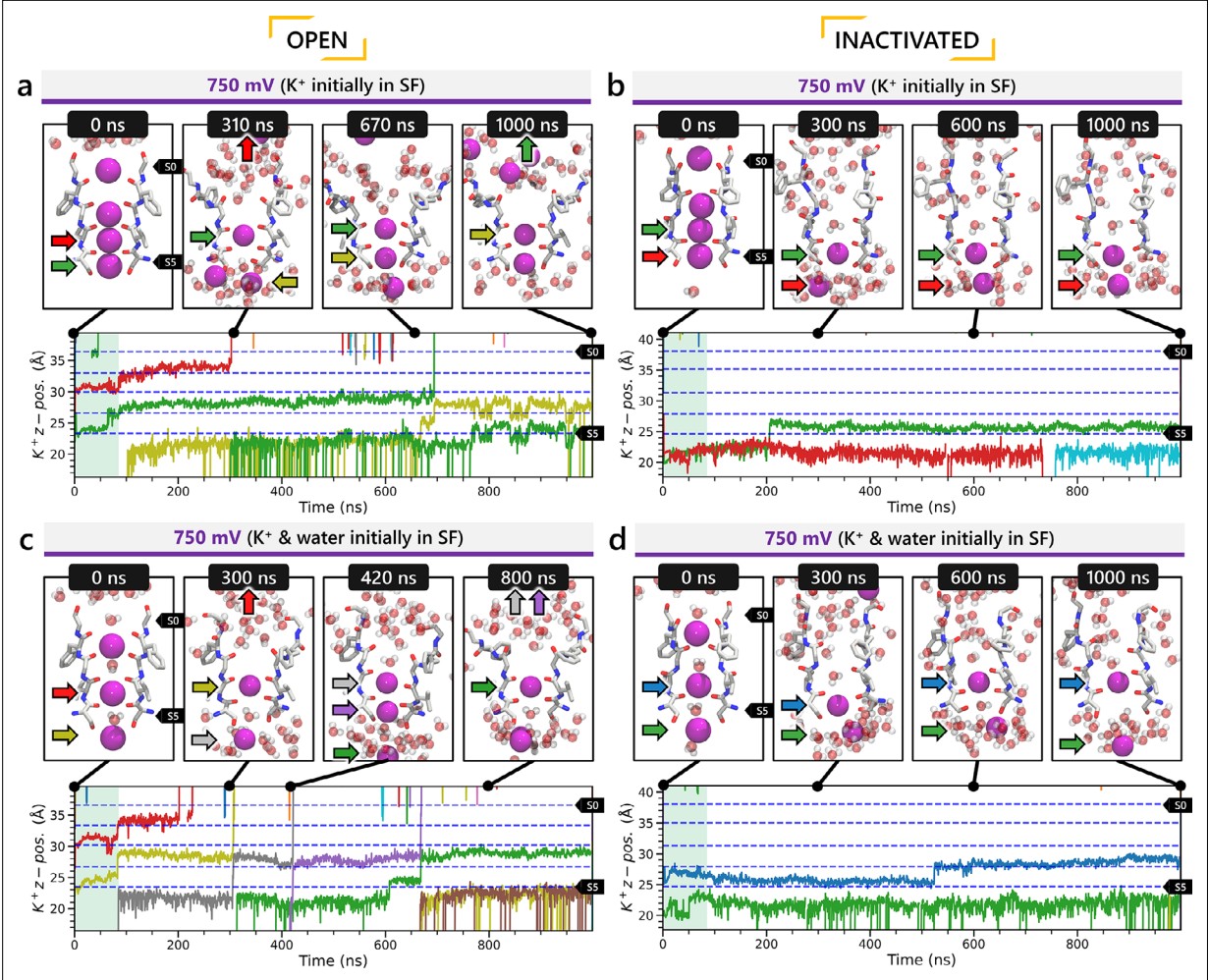

**Figure 5.** Movement of K⁺ ions through hERG selectivity filter (SF) during all-atom molecular dynamics (MD) simulations with the applied membrane voltage. The *z* coordinates of K⁺ ions are tracked as they traverse the pore of the hERG channel from the intracellular gate (lower *y*-axis limit) to the extracellular space (upper *y*-axis limit) under the membrane voltage of 750 mV. Putative K⁺-binding sites in the SF (S0–S5) are marked using blue dashed lines in the plots. Results from MD simulations on the open-state model with the SF occupancy initially configured to have only K⁺ ions (**a**) or alternating K⁺/water molecules (**c**), respectively. Results from MD simulations on the inactivated model with the SF occupancy initially configured to have only K⁺ ions (**b**) or alternating K⁺/water molecules (**d**), respectively.

The online version of this article includes the following figure supplement(s) for figure 5:

**Figure supplement 1.** Setup of MD simulations to assess ion conduction in the open and inactivated hERG channel models.

**Figure supplement 2.** Movement of K⁺ ions through hERG selectivity filter (SF).

**Figure supplement 3.** Analysis of modulations of the selectivity filter (SF) conformations and pore radii over the course of the 1 μs long molecular dynamics (MD) simulations.

**Figure supplement 4.** Analysis of dynamics of the SF and pore conformations over the course of the 1 μs MD simulations.

**Figure supplement 5.** Representative model from the AlphaFold predicted inactivated-state-sampling cluster 3 (AF ic3).

**Figure supplement 6.** Cross-subunit distances between carbonyl oxygens of open-state hERG selectivity filter residues during MD simulations under different applied voltage and initial K⁺ ion position conditions.

**Figure supplement 7.** Sequential dilation steps of hERG upper selectivity filter (SF).

(*Figure 5—figure supplement 2g*) when the SF was initially filled with water molecules and ions, while no conduction events were observed in the remaining cases (*Figure 5—figure supplement 2e, f, h*).

## Conformational changes during MD

Subsequently, we conducted an analysis of pore radius changes throughout the MD simulations (see full results in *Figure 5—figure supplement 3*). In under zero voltage conditions, we observed consistent and distinct pore radius profiles across all simulations within their respective models (left panel in *Figure 5—figure supplement 4a*). Specifically, MD simulations featuring the inactivated-state model consistently displayed a narrower pore radius when compared to simulations involving the open-state model. However, when subjected to high-voltage conditions, the open-state model exhibited a shift toward an inactivated-like state, leading to a reduction in the pore width (right panel in *Figure 5—figure supplement 4a*), which is consistent with an increased hERG channel inactivation propensity at more depolarized voltages (*Vandenberg et al., 2012*).

Although we did not observe the outward flipping of the V625 backbone carbonyl oxygens in the SF during the 1 μs long MD simulations of the open-state model, we did observe the flipping of the F627 backbone carbonyl oxygens as shown in *Figure 5—figure supplement 4b*. Interestingly, this specific SF conformation, with flipped F627 but inward-facing V625 carbonyl oxygens, is also present in Cluster 3 of the AlphaFold-predicted models in *Figure 2*. To explore this further, we investigated the top model from inactivated-state-sampling Cluster 3, which had not been included in prior simulations (*Figure 5—figure supplement 5*). This model features flipped G626 and G628 backbone carbonyls while maintaining an inward-facing V625 carbonyl oxygen conformation (*Figure 5—figure supplement 5a, b*). To evaluate the functional relevance of this SF configuration in the new model, we performed additional MD simulations (two replicates, 1 μs each at 750 mV) with varied initial K$^+$ ion and water arrangements. Both simulations showed multiple K$^+$ conduction events (*Figure 5—figure supplement 5c, d*) for this model, supporting our earlier observation that dilation of the upper SF can still permit ion conduction, provided that residue V625 backbone carbonyls remain inward-facing. As a result, we named this model Open (AlphaFold inactivated-state-sampling Cluster 3, or AF ic3) to differentiate it from the Open (PDB 5VA2-based) model.

These findings further highlight the critical role of V625 in regulating ion conduction through the SF of the hERG channel. In the inactivated-state model simulations, elevated membrane voltage increased the likelihood of V625 backbone carbonyls adopting a conductive orientation (inward-facing). However, even a single outward-facing V625 carbonyl oxygen was sufficient to block K$^+$ conduction through the SF.

## Comparison with previously reported K$^+$ channel C-type inactivation mechanisms

Cuello et al. in their study of KcsA channel identified a similar constriction at G77 within the SF and a corresponding reorientation of the V76 carbonyl, resulting in a dilation in the SF at this location and corresponding loss of the S2 and S3 ion-binding sites (*Cuello et al., 2010*). They suggested this backbone rearrangement as a fundamental molecular mechanism underlying C-type inactivation in K$^+$ channels (*Cuello et al., 2010*). In other studies on Shaker and K$_v$1.3 channels, dilation in the upper SF that disrupts the S1 and S2 K$^+$-binding sites has been proposed to be a potential C-type inactivation mechanism (*Chandy et al., 2023*; *Selvakumar et al., 2022*; *Tan et al., 2022*; *Tyagi et al., 2022*). Similar dilations in the SF are also predicted by AlphaFold2, particularly within Cluster 3 of the predicted inactivated-state hERG channel clusters shown in *Figure 2a*. Although these models were not simulated under our study, such dilated conformations of the SF also emerged during our MD simulations of the open-state model under applied voltage.

We further compared SF conformations by plotting the cross-subunit distances between the carbonyl oxygen atoms of SF residues of open-state hERG channel MD simulations at 750 and 500 mV applied voltages, as shown in *Figure 5—figure supplement 6*. The dilation observed in the hERG channel, which also occurs in the upper SF, differs from that in the aforementioned K$^+$ channels. In Shaker-family channels, the most considerable widening occurs at the SF tyrosine residue (Y445 in Shaker/Y377 in K$_v$1.2) immediately below the topmost SF residue (G446 in Shaker/G378 in K$_v$1.2) (*Tan et al., 2022*; *Wu et al., 2025*). Conversely, in the hERG channel, the topmost SF residue (G628) exhibits the most significant widening, followed by the residue immediately below it (F627). Our MD

simulations of the hERG channel reveal that its dilation process involves two sequential steps: SF near residues F627 dilates first, followed by SF near topmost G628 residues. The latter step occurs faster at higher voltages (750 mV) compared to lower voltages (500 mV). We present these steps in *Figure 5—figure supplement 7*. Notably, despite the dilation of the hERG SF, ion conduction is still observed across all replicas, in contrast to the Shaker channel (*Chandy et al., 2023*; *Selvakumar et al., 2022*; *Tan et al., 2022*; *Tyagi et al., 2022*).

## Computational drug docking reveals state-specific differences in drug-binding affinities

We utilized Rosetta GALigandDock software (*Park et al., 2021*) to dock 19 drugs from different classes, considering their multiple protonation states, into our hERG state-specific channel models. This process aimed to evaluate and corroborate state-dependent binding interactions with experimental studies, specifically in terms of relative binding affinities. *Figure 6—figure supplement 1* presents these findings in the form of Rosetta GALigandDock (*Park et al., 2021*) binding energies (lower, more negative values mean more favorable binding). Consistent with published studies, most drugs showed stronger binding to the inactivated-state hERG channel model, including astemizole, terfenadine, cisapride, d/l-sotalol, dofetilide (*Ficker et al., 1998*; *Kamiya et al., 2008*; *Perrin et al., 2008*), haloperidol (*Suessbrich et al., 1997*), and E-4031 (*Numaguchi et al., 2000*; *Wang et al., 1997*). Drugs like moxifloxacin (*Alexandrou et al., 2006*), quinidine (*Perrin et al., 2008*), verapamil (*Duan et al., 2007*), and perhexiline (*Perrin et al., 2008*) did not show strong preference for the inactivated-state model, aligning with findings from hERG experimental studies using inactivation-deficient mutants (*Perrin et al., 2008*) or 'step-ramp' voltage protocol (*Alexandrou et al., 2006*).

As a control, we also included docking results for the presumed open-state model from inactivated-state-sampling Cluster 3 (referred to as Open, AF ic3). Although its SF differs from both the experimental open and predicted inactivated-state models, our previous simulations confirmed that it supports ion conduction. Structurally, its pore most closely resembles the open state (*Figure 5—figure supplement 5b*) with only minor differences, and accordingly, its drug docking profile aligns well with that of the open-state model. These results further support the interpretation that the Cluster 3 model represents an alternative open-state conformation. *Table 3* provides an overview of all models examined, along with qualitative insights into their observed behaviors thus far.

In our GALigandDock docking results, most drugs exhibited increased binding affinity to the closed-state hERG channel model compared to the open-state hERG channel model. Drugs are unable to bind to the closed state from the intracellular space because the pore is closed. However, they can become trapped if they are already bound when the channel transitions from an open to a closed state, as shown in experiments for dofetilide (*Windley et al., 2017*), cisapride (*Windley et al., 2017*), terfenadine (*Windisch et al., 2011*; *Windley et al., 2017*), E-4031 (*Windisch et al., 2011*), and nifekalant (*Kamiya et al., 2006*).

To model drug trapping, we placed the drug in a pocket beneath the SF in the closed pore configuration before docking. However, this method does not consider how the conformational shift from

**Table 3.** Overview of models and qualitative observations.

| Model | Origin | K⁺ ion conduction | Qualitative drug-binding trend |
|---|---|---|---|
| Open (5VA2) | Based on hERG cryo-EM structure PDB 5VA2 (*Wang and MacKinnon, 2017*) with loops rebuilt using Rosetta | Yes | Low affinity for drugs known to bind preferentially to the inactivated state |
| Inactivated | Top model from Cluster 2 in AlphaFold inactivated-state-sampling attempt | No | High affinity for drugs known to bind preferentially to the inactivated state |
| Closed | Top model from Cluster 1 in AlphaFold closed-state-sampling attempt | Not tested (pore is closed) | High affinity for most drugs, assuming pore closure does not eject bound compounds |
| Open (AF ic3) | Top model from Cluster 3 in AlphaFold inactivated-state-sampling attempt | Yes | Low affinity for drugs known to bind preferentially to the inactivated state |
| Open (AF control) | Top model from Cluster 1 (only cluster) in AlphaFold open-state-sampling attempt. Produced solely as a control to show that AlphaFold could reproduce complete models resembling PDB 5VA2. | Not tested as the model is structurally identical to Open (5VA2) | Not tested as the model is structurally identical to Open (5VA2) |

the open to the closed state might influence drug binding. Under physiological conditions, the pore gating motion from open to closed might expel drugs from the pore instead of pushing them deeper. This limitation might account for some inconsistencies noted in our docking study, particularly regarding the apparent trapping of drugs such as amiodarone and haloperidol, which is at odds with experimental results (*Stork et al., 2007*). However, these preliminary results could pave the way for more thorough investigations, employing advanced computational techniques to delve deeper into the dynamics of drug trapping (*Branduardi and Faraldo-Gómez, 2013*; *Miao et al., 2020*).

## State-specific molecular determinants of hERG channel block by terfenadine, dofetilide, moxifloxacin, astemizole, and E-4031

*Figure 6* highlights the binding profiles of terfenadine, dofetilide, and moxifloxacin. Terfenadine and dofetilide are modeled in their cationic forms, while moxifloxacin is in its zwitterionic form. Experimental evidence (*Ficker et al., 1998*; *Kamiya et al., 2008*; *Perrin et al., 2008*) indicates that terfenadine and dofetilide preferentially bind to the inactivated state of hERG, whereas moxifloxacin does not show this state-specific preference. Notably, both terfenadine and dofetilide have been associated with TdP arrhythmia and have been withdrawn or restricted in clinical use, while moxifloxacin is generally considered safer (*Alexandrou et al., 2006*; *Jaiswal and Goldbarg, 2014*; *Monahan et al., 1990*; *Orvos et al., 2019*; *Yang et al., 2020*). Here, we investigate whether molecular differences in state-dependent binding modes, particularly to the inactivated state, and corresponding differences in binding affinities may help explain their varying proarrhythmic risks.

### Terfenadine (*Figure 6a*)

- In the PDB 5VA2-derived open-state model, terfenadine forms strong π–π stacking interactions with the phenol side chains of Y652 (for 3 subunits), anchoring its aromatic rings just below the Y652 ring plane. Y652 and S660 engage in hydrogen bonding with terfenadine, while F656 contributes a hydrophobic contact via its backbone, further stabilizing the ligand within the central cavity.
- In the AF ic3 open-state model, terfenadine adopts a more vertical orientation. It forms π–π stacking with the phenol ring of Y652 and engages in hydrogen bonding with residue S660. The binding pose is further supported by hydrophobic contacts with residues T623, Y652 (on 2 subunits), and S660.
- In the inactivated-state model, terfenadine binds much deeper in the pore and forms a broader array of interactions. It engages in π–π stacking with Y652 and F557, while its hydroxyl group forms hydrogen bonds with residues L622, S624, and S649. Additional hydrophobic contacts occur with L622, T623, S649, M651, and F656, creating a tightly packed interaction network.
- In the closed-state model, terfenadine becomes further embedded in the pore. Two F656 residues form π–π stacking interactions with its phenol ring, and the ligand is stabilized by hydrophobic interactions with residues S621, L622, T623, S624, M645, S649, Y652, and additional F656 residues.
- Supporting our findings, Kamiya et al. demonstrated that alanine substitutions at T623, S624, Y652, and F656 significantly reduced the sensitivity of hERG to block by terfenadine (*Kamiya et al., 2006*). In addition, Saxena et al. reported that F557L and Y652A mutations significantly reduced terfenadine-induced hERG inhibition (*Saxena et al., 2016*).

### Dofetilide (*Figure 6b*)

- In the PDB 5VA2-derived open-state model, dofetilide predominantly forms polar interactions, with hydrogen bonds involving residues S660 and A653 across multiple subunits, and hydrophobic contacts with residue G657 contributing to its stabilization within the central cavity.
- In the AF ic3 open-state model, dofetilide binds slightly deeper and adopts a more upright orientation. It forms π–π stacking interactions with the phenol side chains of Y652 (of 2 subunits), anchoring its aromatic core. It also engages in hydrogen bonding with Y652 and hydrophobic contacts with residues Y652, S660, and F656.
- In the inactivated-state model, dofetilide engages in its most extensive interaction network. It binds deep in the pore, forming π–π stacking with Y652 and forming hydrogen bonds with residues T623 and S649. Additional polar contacts are observed with Y652 and T623, while hydrophobic stabilization is provided by contacts with residues S624, S649, M554, and F557. This

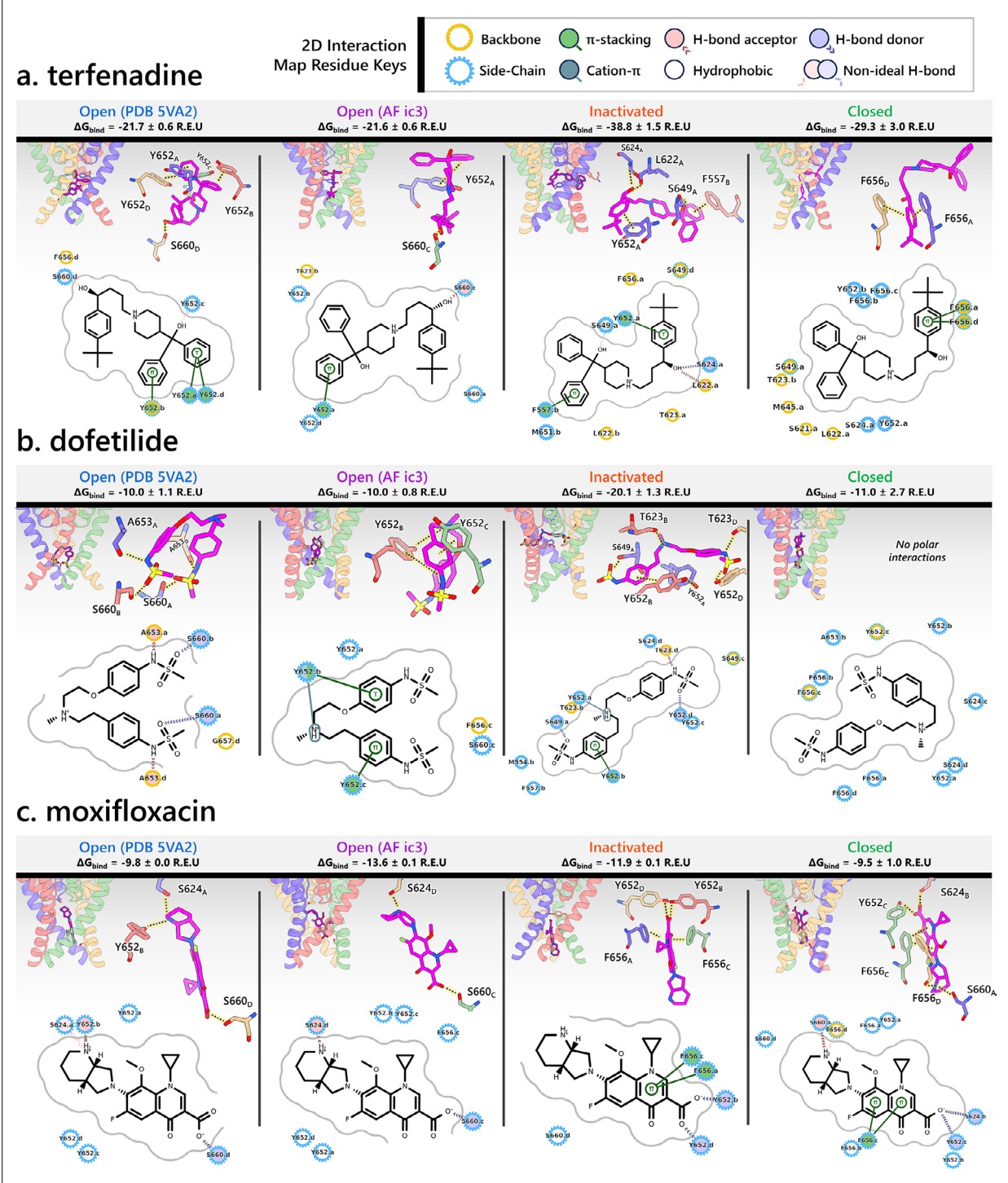

**Figure 6.** Visualization of interactions for terfenadine (**a**), dofetilide (**b**), and moxifloxacin (**c**) with different hERG channel models. Each panel includes four subpanels showcasing drug interactions with the open- (PDB 5VA2-derived and AlphaFold-predicted from inactivated-state-sampling Cluster 3, i.e., AF ic3), inactivated-, and closed-state hERG channel models. The estimated drug-binding free energies, $\Delta G_{bind}$, are given in Rosetta energy units (R.E.U.) and shown as averages ± standard deviations. In each subpanel, an overview of where the drug binds within the hERG channel pore is shown on the upper left, a 3D visualization of interactions between each channel residue (blue, red, green, and tan colored residues are from the subunit A, B, C, or D, respectively) to the drug (magenta) is shown on the upper right, and a 2D ligand–protein interaction map is shown at the bottom. A continuous gray line depicts the contour of the protein-binding site, and any breaks in this line indicate areas where the ligand is exposed to the solvent.

The online version of this article includes the following figure supplement(s) for figure 6:

*Figure 6 continued on next page*

Figure 6 continued

**Figure supplement 1.** GALigandDock drug docking free energies for different hERG channel models.

**Figure supplement 2.** Astemizole (a) and E-4031 (b) binding to different hERG channel models and cryo-EM structures.

comprehensive interaction profile reflects the experimentally observed preference of dofetilide for the inactivated state (*Perrin et al., 2008*), which might contribute to its increased proarrhythmic risk (*Ficker et al., 1998*; *Jaiswal and Goldbarg, 2014*; *Yang et al., 2020*).

- In the closed-state model, dofetilide is positioned even deeper in the pore, likely retained by the narrowed cavity. Though it lacks strong polar contacts in this state, the surrounding residues, including S624, Y652, A653, and F656, encase the ligand and contribute to its stabilization via hydrophobic interactions.
- Consistent with our docking results, experimental data support the involvement of these residues in dofetilide binding: Saxena et al. reported reduced binding following F557L and M554A mutations (*Saxena et al., 2016*); Lees-Miller et al. showed that the F656V mutation weakens dofetilide block (*Lees-Miller et al., 2000a*). Kamiya et al.'s alanine-scanning mutagenesis identified T623A, S624A, Y652A, and F656A as significantly reducing dofetilide potency, along with spatially adjacent residues G648A and V659A (*Kamiya et al., 2006*). Additional work by Stepanovic et al. demonstrated that residue A653 mutations also alter hERG block by dofetilide (*Stepanovic et al., 2009*).

## Moxifloxacin (*Figure 6c*)

- Across all the models, the geometry of moxifloxacin favors a vertically oriented binding pose within the hERG channel pore, with minimal bending of the molecule.
- In the PDB 5VA2-derived open-state model, moxifloxacin reaches deep into the pore, where it forms hydrogen bonds with residues S624 and Y652 via the secondary ammonium group. The carboxylate oxygen engages in hydrogen bonding with residue S660.
- The AF ic3 open-state model shows a similar deep binding pose, stabilized primarily through hydrogen bonds with S624 and S660.
- In contrast, the inactivated-state model reveals a shallower binding position. Despite this, moxifloxacin is stabilized through multiple π–π stacking interactions between its quinolone ring system and F656 residues from opposing subunits, along with hydrogen bonds involving its carboxylate group and Y652 side chains from multiple subunits. Unlike terfenadine and dofetilide, moxifloxacin does not show enhanced binding in the inactivated state.
- The closed-state model similarly features π–π stacking with F656 residues and a network of hydrogen bonds involving S660, Y652, and S624, effectively enclosing the ligand.
- Previous studies reported that mutation at Y652 significantly reduced the sensitivity of hERG channel inhibition by moxifloxacin (*Alexandrou et al., 2006*; *Shinozawa et al., 2017*), consistent with our observation that Y652 plays a central role in stabilizing moxifloxacin binding across all the states we tested. Compared to other drugs, hERG residue F656 in this case only engages in π-stacking in two channel states and appears less essential for binding as moxifloxacin can be anchored through other polar and hydrophobic contacts, thus explaining the limited effect of the F656 mutation (*Alexandrou et al., 2006*).

Recently cryo-EM structures of the hERG channel in complex with astemizole and E-4031 were reported (*Miyashita et al., 2024*). Here, we compare drug binding in our open- and inactivated-state models, using the cationic forms of astemizole and E-4031, with the corresponding experimental structures (*Figure 6—figure supplement 2*). Binding in the closed state is excluded as the pore architecture deviates too much from those in the cryo-EM structures. Experimental data (*Perrin et al., 2008*) indicate that both astemizole and E-4031 bind more potently to the inactivated state.

## Astemizole (*Figure 6—figure supplement 2a*)

- In the PDB 5VA2-derived open-state model, astemizole binds centrally within the pore cavity, adopting a bent conformation that allows both aromatic ends of the molecule to engage in π–π stacking with the side chains of Y652 from two opposing subunits. Hydrophobic contacts are observed with S649 and F656.
- In the AF ic3 open-state model, the ligand is stabilized through multiple π–π stacking interactions with Y652 residues from three subunits, forming a tight aromatic cage around its triazine

and benzimidazole rings. Hydrophobic interactions are observed with hERG residues T623, S624, Y652, F656, and S660.

- In the inactivated-state model, astemizole adopts a compact, horizontally oriented pose deeper in the channel pore, forming the most extensive interaction network among all the states. The ligand is tightly stabilized by multiple π–π stacking interactions with Y652 residues across three subunits and forms hydrogen bonds with residues S624 and Y652. Additional hydrophobic contacts are observed with residues F557, L622, S649, and Y652.
- Consistent with our findings, the electrophysiology study by Saxena et al. identified hERG residues F557 and Y652 as crucial for astemizole binding, as determined through mutagenesis (*Saxena et al., 2016*).
- In the cryo-EM structure (PDB 8ZYO) (*Miyashita et al., 2024*), astemizole is stabilized by π–π stacking with Y652 residues. However, no hydrogen bonds are detected, which may reflect limitations in cryo-EM resolution rather than true absence of contacts. Additional hydrophobic interactions are observed with L622 and G648.

## E-4031 (*Figure 6—figure supplement 2b*)

- In the PDB 5VA2-derived open-state model, E-4031 binds within the central cavity primarily through polar interactions. It forms a π–π stacking interaction with residue Y652, anchoring one end of the molecule. Polar interactions are observed with residues A653 and S660. Additional hydrophobic contacts are observed with residues A652 and Y652.
- In the AF ic3 open-state model, E-4031 adopts a slightly deeper pose within the central cavity stabilized by dual π–π stacking interactions between its aromatic rings and hERG residues Y652. Additional hydrogen bonds are observed with residues S624 and Y652, and hydrophobic contacts are observed with residues T623 and S624.
- In the inactivated-state model, E-4031 adopts its deepest and most stabilized binding pose, consistent with its experimentally observed preference for this state. The ligand is stabilized by multiple π–π stacking interactions between its aromatic rings and hERG residues Y652 from opposing subunits. The sulfonamide NH group engages in hydrogen bonding with residue S649, while the piperidine nitrogen as well as the central carbonyl oxygen hydrogen bond with residues Y652. Hydrophobic contacts with residues S624 and F656 further reinforce the binding, enclosing the ligand in a densely packed aromatic and polar environment.
- Previous mutagenesis study showed that mutations involving hERG residues F557, T623, S624, Y652, and F656 affect E-4031 binding (*Helliwell et al., 2023*).
- In the cryo-EM structure (PDB 8ZYP) (*Miyashita et al., 2024*), E-4031 engages in a single π–π stacking interaction with hERG residue Y652, anchoring one end of the molecule. The remainder of the ligand is stabilized predominantly through hydrophobic contacts involving residues S621, L622, T623, S624, M645, G648, S649, and additional Y652 side chains, forming a largely non-polar environment around the binding pocket.

In both cryo-EM structures, astemizole and E-4031 adopt binding poses that closely resemble the inactivated-state model in our docking study, consistent with experimental evidence that these drugs preferentially bind to the inactivated state (*Perrin et al., 2008*). This raises the possibility that the cryo-EM structures may capture an inactivated-like channel state. However, closer examination of the SF reveals that the cryo-EM conformations more closely resemble the open-state PDB 5VA2 structure (*Wang and MacKinnon, 2017*), which has been shown to be conductive here and in previous studies (*Miranda et al., 2020*; *Yang et al., 2020*).

The conformational differences between the cryo-EM and open-state docking results may reflect limitations of the docking protocol itself, as GALigandDock assumes a rigid protein backbone and cannot account for ligand-induced shifts. In our open-state models, the hydrophobic pocket beneath the SF is too small to accommodate bulky ligands (*Figure 3a, b*), whereas the cryo-EM structures show a slight outward shift in the S6 helix that expands this space (*Figure 6—figure supplement 2*). These allosteric rearrangements, though small, fall outside the scope of the current docking protocol, which lacks flexibility to capture these local, ligand-induced adjustments (*Harris et al., 2024*).

In contrast, docking to the AlphaFold-predicted inactivated-state model reveals a reorganization beneath the SF that creates a larger cavity, allowing deeper ligand insertion. Notably, neither our inactivated-state docking nor the available cryo-EM structures show strong interactions with residues F656. However, in the AlphaFold-predicted inactivated-state model, the more extensive protrusion of F656 into the central cavity may further occlude the drug's egress pathway, potentially trapping the

ligand more effectively. This could explain why the mutation of F656 significantly reduces the binding affinity of E-4031 (*Helliwell et al., 2023*). These findings suggest that inactivation may trigger a series of modular structural rearrangements that influence drug access and binding affinity, with different aspects potentially captured in various computational and experimental studies, rather than resulting from a single, uniform conformational change.

## Validation of state-dependent drug block with experimental data using hERG Markov model

There are several complications that make it difficult to directly compare experimental binding affinities with predicted affinities from simulations. During electrophysiological recordings of hERG inhibition by various drugs, the hERG channel has been shown to adopt various functional states, presumably corresponding to protein conformation states. These states can be bound by drugs with varying affinities, with drug ionization state also being a contributing factor. Additionally, the variability in experimental protocols affects the measured affinities (*Gomis-Tena et al., 2020*). In general, electrophysiological measurements report the $IC_{50}$, the drug concentration required for 50% inhibition of current. However, the $IC_{50}$ value is not directly comparable to computed affinities from drug docking.

To address these challenges, we developed a novel computational approach that combines modeling and simulation to predict hERG channel conformational state probabilities (open, closed, and inactivated) over time. First, we collected a comprehensive set of experimental data and employed a hERG functional model with five functional states, which was extensively validated in our earlier study (*Romero et al., 2015*). For each drug, we ran in silico electrophysiological experiments under the same conditions as the experimental studies, allowing us to calculate the relative probabilities of the various hERG channel states specific to the drug and protocol. These state probabilities were then used to refine the computed binding affinities from docking simulations. We adjusted the affinities for both neutral and charged forms of each drug according to their prevalence in each conformational state. This method allowed us to scale the binding predictions based on the likelihood of each channel state occurring during the experimental protocols. Finally, we compared the simulated binding affinities with experimental hERG drug potencies (*Table 4*), offering a new validation technique that enhances the accuracy of our predictions and helps reconcile the differences between experimental $IC_{50}$ measurements and computed affinities.

We compared the experimental drug potencies with the simulated binding affinities, starting with the traditional approach of using only open-state docking simulations (*Figure 7a, c*), commonly employed in ion channel pharmacology due to a scarcity of multi-state models, and then extended our analysis to include drug binding to different states (*Figure 7b, d*). Using only the open-state model (PDB 5VA2) yielded a moderate correlation with experimental data ($R^2 = 0.43$, $r = 0.66$, *Figure 7a*). Incorporating multi-state binding (weighted by their experimental distributions) improved the correlation substantially ($R^2 = 0.63$, $r = 0.79$, *Figure 7b*), boosting predictive power by 47% and underscoring the value of multi-state modeling. Importantly, this improvement was achieved without considering potential drug-induced allosteric effects on hERG channel conformation and gating, which will be addressed in future work.

Next, we substituted the PDB 5VA2-based open-state model with the AF ic3 open-state model. Docking to this alternative model alone produced similar performance ($R^2 = 0.44$, $r = 0.66$, *Figure 7c*), and incorporating it into the multi-state ensemble further improved the correlation with experiments ($R^2 = 0.64$, $r = 0.80$, *Figure 7d*), representing a 45% gain in $R^2$ and matching the performance of multi-state docking results based on the PDB 5VA2-derived model.

These findings suggest that the predictive power of computational drug docking is enhanced not merely by the accuracy of individual models, but by the structural diversity and complementarity provided by an ensemble of conformations. Rather than relying solely on a single experimentally determined structure, the ensemble benefits from incorporating AlphaFold-predicted models that capture alternative conformations identified through our state-specific sampling approach. These diverse models reflect different structural features, which together offer a more comprehensive representation of the channel's binding landscape and enhance the predictive performance of computational drug docking. Overall, these results reinforce that multi-state modeling offers a more realistic and predictive framework for understanding drug-channel interactions than traditional single-state approaches, emphasizing the value of both individual model evaluation and their collective integration.

**Table 4.** Data used for validating binding affinities from hERG channel-drug docking simulations with experiments.

| Drugs | Ionization states and prevalence at pH 7.4 | Simulated binding affinities (Rosetta energy units) | | | | hERG channel state distribution when the tail current was observed | | | $\Delta G_{bind,\,sim}$ (Rosetta energy units) | $\Delta G_{pot,\,exp}$ (kcal/mol) | $IC_{50}$ (nM) | Studies referenced |
|---|---|---|---|---|---|---|---|---|---|---|---|---|
| | | Open (5VA2) | Open (AF ic3) | Inact. | Closed | Open | Inact. | Closed | | | | |
| Astemizole | Neutral 1.50%* | −13.84 ± 1.15 | −14.49 ±0.23 | −26.33 ±0.82 | −22.61 ±2.34 | | | | | | | *Chiu et al., 2004* |
| | Cationic 98.50%* | −19.79 ± 1.55 | −15.12 ±1.16 | −32.51 ±0.83 | −25.63 ±1.76 | 43.96% | 52.43% | 3.61% | −26.58 ±0.80 | −10.76 | 26 | |
| Terfenadine | Neutral 1.50% | −21.55 ± 0.80 | −22.43 ± 1.61 | −31.51 ±0.01 | −30.56 ±0.35 | | | | −30.94 ±1.22; | | | *Orvos et al., 2019; Tanaka et al., 2014* |
| | Cationic 98.50% | −21.73 ± 0.56 | −21.56 ± 1.47 | −38.80 ±2.24 | −29.25 ±2.98 | 45.21%; 57.91% | 53.93%; 40.71% | 0.86%; 1.38% | −28.74 ±0.95 | −10.65; −10.66 | 31; 30.60 | |
| Cisapride | Neutral 22.40% | −11.23 ± 1.10 | −15.72 ± 0.23 | −21.44 ±1.53 | −18.92 ±0.75 | | | | | | | *Rampe et al., 1997; Chiu et al., 2004* |
| | Cationic 77.60% | −15.19 ± 0.90 | −15.08 ± 0.84 | −30.43 ±1.09 | −22.35 ±0.76 | 41.89%; 43.96% | 51.15%; 52.43% | 6.96%; 3.61% | −22.03 ±0.56; −21.97 ±0.58 | −9.92; −11.58 | 44.5; 6.90 | |
| Verapamil | Neutral 0.60% | −17.52 ± 1.40 | −14.14 ± 0.83 | −16.68 ±0.22 | −21.12 ±1.71 | | | | | | | *Zhang et al., 1999; Johnson and Trudeau, 2023* |
| | Cationic 99.40% | −20.54 ± 3.84 | −15.12 ± 0.23 | −20.01 ±1.09 | −25.10 ±5.17 | 56.98%; 9.00% | 40.07%; 90.62% | 2.94%; 0.38% | −20.44 ±2.22; −20.06 ±1.04 | −9.24; −9.57 | 143; 180.40 | |
| d-Sotalol | Neutral 0.62%* | −6.20 ± 0.06 | −6.03 ±0.44 | −16.20 ±2.08 | −8.36 ±0.44 | | | | | | | *Perrin et al., 2008* |
| | Cationic 99.38%* | −5.83 ± 1.46 | −6.90 ±0.58 | −15.04 ±1.28 | −10.91 ±1.17 | 91.54% | 4.50% | 3.96% | −6.45 ±1.33 | −4.66 | 515,500 | |
| l-Sotalol | Neutral 0.62%* | −6.59 ± 0.38 | −7.44 ± 0.30 | −16.41 ±0.88 | −10.42 ±0.71 | | | | | | | *Perrin et al., 2008* |
| | Cationic 99.38%* | −6.31 ± 0.98 | −7.32 ± 1.21 | −16.64 ±0.27 | −11.48 ±1.10 | 91.54% | 4.50% | 3.96% | −6.98 ±0.89 | −4.66 | 515,500 | |
| Dofetilide | Neutral 5.70% | −7.30 ± 0.09 | −3.20 ± 0.73 | −10.51 ±2.24 | −6.43 ±1.67 | | | | −17.38 ±1.00; | | | *Ficker et al., 1998; Li et al., 2012* |
| | Cationic 94.30% | −10.04 ± 1.11 | −10.08 ±0.75 | −20.05 ±1.32 | −11.00 ±2.67 | 19.23%; 57.04% | 77.68%; 40.14% | 3.08%; 2.83% | −13.77 ±0.78 | −8.77; −10.46 | 320; 17.90 | |
| Haloperidol | Neutral 15.50% | −11.02 ± 0.06 | −11.76 ± 0.04 | −20.85 ±0.18 | −19.72 ±0.84 | | | | | | | *Suessbrich et al., 1997* |
| | Cationic 84.50% | −12.84 ± 1.45 | −13.05 ± 0.02 | −26.34 ±0.13 | −21.80±0.70 | 54.25% | 13.37% | 32.38% | −17.17 ±0.69 | −8.10 | 1,000 | |
| Amiodarone | Neutral 2.00% | −10.76 ± 1.38 | −16.87 ± 0.43 | −27.10 ±0.98 | −18.00 ±0.79 | | | | | | | *Zhang et al., 2016* |
| | Cationic 98.00% | −13.38 ± 1.95 | −17.48 ± 1.21 | −30.49 ±1.12 | −23.95 ±1.55 | 45.21% | 53.93% | 0.86% | −22.64 ±1.05 | −10.42 ±0.07 | 45±5.20 | |
| E-4031 | Neutral 21.09%* | −9.87 ± 0.17 | −9.73 ± 0.24 | −17.27 ±1.92 | −11.54 ±2.45 | | | | | | | *Zhou et al., 1998* |
| | Cationic 78.91%* | −11.55 ± 0.60 | −14.57 ± 1.64 | −24.76 ±0.59 | −11.96 ±1.78 | 57.70% | 40.54% | 1.76% | −16.07 ±0.37 | −11.51 | 7.7 | |
| Clozapine | Neutral 100%* | −8.62 ±0.45 | −8.20 ± 0.29 | −13.96 ±0.41 | −16.84 ±0.99 | 62.12%; 67.97% | 25.62%; 27.99% | 12.27%; 4.05% | −11.00 ±0.32; −10.45 ±0.33 | −6.14; −6.26 | 28,300; 22,900 | *Lee et al., 2006* |
| NS1643 | Neutral 100%* | −13.72 ± 1.97 | −13.89 ± 0.65 | −19.44 ±0.42 | −21.73 ±0.27 | 69.21% | 28.79% | 2.00% | −15.53 ±1.37 | −7.06 | 10,500 | *Hansen et al., 2006* |

*Table 4 continued on next page*

*Table 4 continued*

| Drugs | Ionization states and prevalence at pH 7.4 | Simulated binding affinities (Rosetta energy units) | | | | hERG channel state distribution when the tail current was observed | | | ΔG$_{bind, sim}$ (Rosetta energy units) | ΔG$_{pot, exp}$ (kcal/mol) | IC$_{50}$ (nM) | Studies referenced |
|---|---|---|---|---|---|---|---|---|---|---|---|---|
| | | Open (5VA2) | Open (AF ic3) | Inact. | Closed | Open | Inact. | Closed | | | | |
| Moxifloxacin | Neutral 0% | − 8.27 ± 0.54 | − 9.82 ± 0.34 | −10.00 ±0.27 | −19.85 ±0.84 | | | | −10.62 ±0.23; −10.94 ±0.07 | −5.65 ±0.04; −6.31 | 65,000±4,200; 35,700 | *Alexandrou et al., 2006; Chen et al., 2005* |
| | Zwitterionic 100%* | − 9.78 ± 0.03 | − 13.62 ± 0.08 | −11.93 ±0.12 | −9.45 ±0.99 | 34.82%; 45.21% | 42.72%; 53.93% | 22.46%; 0.86% | | | | |
| Quinidine | Neutral 2.20% | − 7.66 ± 0.64 | − 10.89 ± 0.41 | −12.20 ±0.07 | −17.54 ±1.53 | | | | −12.33 ±0.28; −11.77 ±0.22 | −8.23 ±0.07; −9.06 | 800±100; 410 | *Yan et al., 2016; Paul et al., 2002* |
| | Cationic 97.70% | − 9.49 ± 0.17 | − 14.32 ± 1.41 | −12.87 ±0.33 | −20.28 ±1.32 | 57.17%; 32.84% | 23.59%; 66.68% | 19.24%; 0.48% | | | | |
| Perhexiline | Neutral 0% | − 10.47 ± 0.22 | − 12.33 ± 0.72 | −11.66 ±0.74 | −24.39 ±0.56 | | | | −16.19 ±1.26 | −6.89 | 7,800 | *Walker et al., 1999* |
| | Cationic 100% | − 11.20 ± 0.70 | − 12.46 ± 0.02 | −19.05 ±2.04 | −23.90 ±0.66 | 37.57% | 60.59% | 1.84% | | | | |
| Nifekalant | Neutral 3.80% | − 9.68 ± 1.47 | − 12.28 ± 0.29 | −15.40 ±0.95 | −14.89 ±1.56 | | | | −14.23 ±0.99 | −6.89 | 7,900 | *Kushida et al., 2002* |
| | Cationic 96.20% | − 10.93 ± 1.45 | −13.46 ±0.05 | −22.74 ±0.99 | −13.52 ±3.39 | 67.65% | 27.91% | 4.44% | | | | |

Simulated binding affinities are shown as averages ± standard deviations. In the *Ionization states* column, the microspecies distribution percentages were calculated using the ChemAxon software suite and its computed p$K_a$ values (*Pirok et al., 2006*). The asterisk (*) indicates that the value has been adjusted by a few percentage points to account for species with alternative ionization states that were not tested. Cells with values separated by a semicolon (;) and shown by different colors represent data from different studies, listed in the 'studies referenced' column in the same order.

## Discussion
### AlphaFold2 predicts physiologically relevant hERG channel states

In this study, we introduced a methodology to extend AlphaFold2's predictive capabilities by guiding it to sample multiple protein conformations reminiscent of different ion channel states, using the hERG potassium channel as a proof of concept. While other studies have focused on generating diverse protein conformations using AlphaFold (*Del Alamo et al., 2022*; *Kalakoti and Wallner, 2024*; *Lidbrink et al., 2024*; *Sala et al., 2023*), we aimed to take this further by validating the physiological relevance of these predicted states for ion channels through computational simulations and experimental data. By incorporating multiple structural templates and refined input parameters, we directed AlphaFold2 to predict distinct functional states, including the closed and inactivated forms of hERG. This approach is significant for pharmacology, particularly for the hERG channel, an anti-target notorious for state-specific drug interactions linked to cardiotoxicity (*Li and Ramos, 2017*; *Perrin et al., 2008*; *Sanguinetti and Tristani-Firouzi, 2006*; *Vandenberg et al., 2012*; *Yang et al., 2020*).

We employed two complementary strategies to guide AlphaFold2 in predicting physiologically relevant hERG channel conformations. In the first, we used structural fragments representative of specific functional states as templates, prompting AlphaFold2 to reconstruct the full hERG channel accordingly, successfully yielding models with features characteristic of the closed state, such as a constricted pore and deactivated voltage-sensing domains. In the second, we allowed AlphaFold2 to explore a broader conformational space, particularly useful in scenarios where structural knowledge is limited. Remarkably, AlphaFold2 managed to generate conformations that had been previously documented in the literature but not included in the training dataset, showcasing its broad predictive capacity. Among the most compelling findings is the strong correspondence between our AlphaFold2-predicted inactivated-state conformations and those previously proposed through experimental and simulation studies (*Lau et al., 2024*; *Li et al., 2021b*). These structures were identifiable in discrete protein clusters (*Figure 2a*) with high prediction confidence metrics and likely together constitute components of the hERG inactivation mechanism. In this way, AlphaFold2 provided a strong method to reconcile apparently disparate previous experimental and computational data.

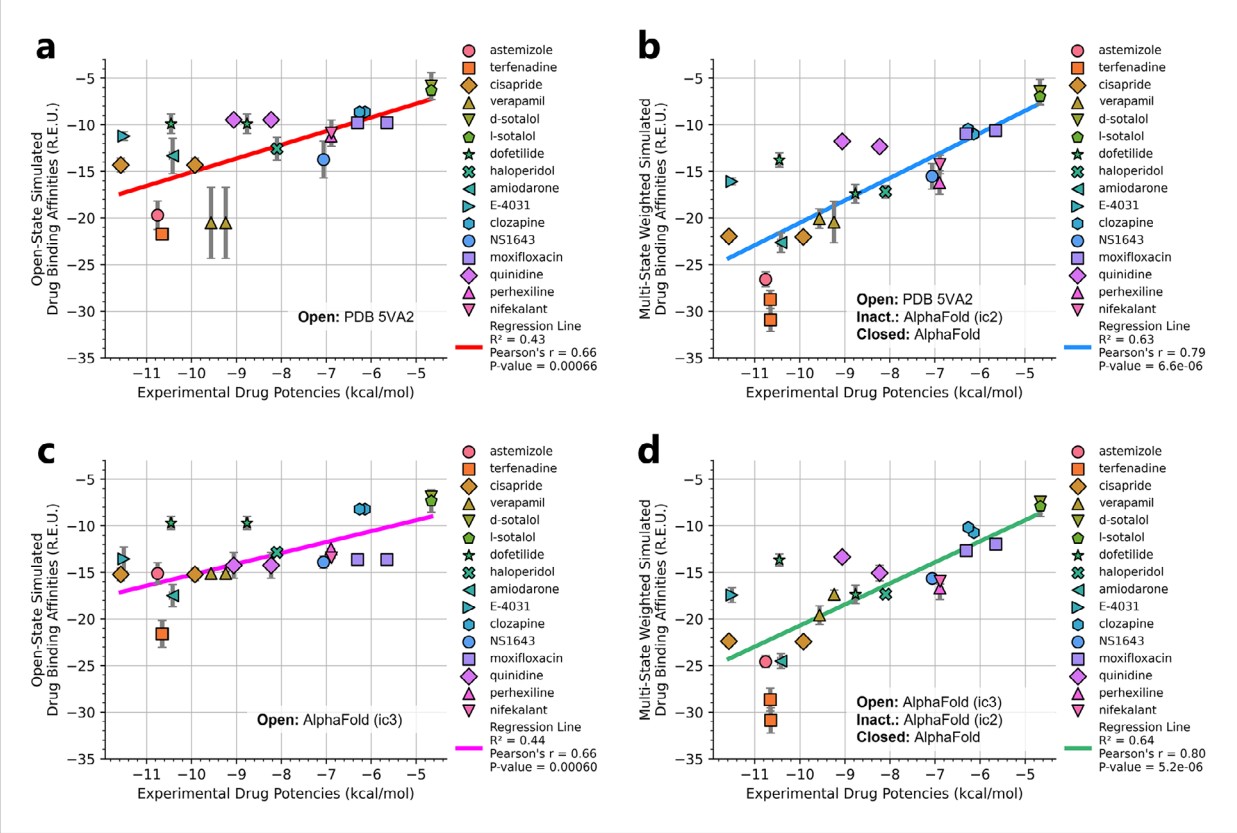

**Figure 7.** Correlation of simulated hERG drug-binding affinities with experimental drug potencies under different modeling scenarios. Single- and multi-state simulated drug-binding affinities (in Rosetta energy units, R.E.U.) are plotted against experimental drug potencies ($IC_{50}$ converted to free energies in kcal/mol). Lower (more negative) values indicate stronger binding. Horizontal error bars reflect uncertainty from experimental $IC_{50}$ measurements, while vertical bars reflect standard deviations in simulated drug-binding affinities ($n = 100$), propagated across ionization states and channel state distributions. A total of 23 measurements representing 16 unique drugs were analyzed. Linear regression was performed using the least-squares method, and exact values for Pearson's $r$, $R^2$, and p-values are reported within the figure. (**a**) Single-state docking using the experimentally derived open-state structure (PDB 5VA2) yields a moderate correlation (the coefficient of determination $R^2 = 0.43$, Pearson correlation coefficient $r = 0.66$). (**b**) Multi-state docking incorporating open (PDB 5VA2), inactivated, and closed-state conformations weighted by experimentally observed state distributions further improve the correlation ($R^2 = 0.63$, $r = 0.79$). (**c**) Single-state docking using an alternative AlphaFold-predicted open state (inactivated-state-sampling Cluster 3, ic3) ($R^2 = 0.44$, $r = 0.66$). (**d**) Multi-state docking combining the AlphaFold-predicted open- (inactivated-state-sampling Cluster 3, ic3), inactivated-, and closed-state models also results in a notable improvement ($R^2 = 0.64$, $r = 0.80$) compared to single-state docking in panel (**c**) and comparable performance to multi-state docking in panel (**b**). These results highlight the enhanced predictive power of multi-state modeling and suggest that structural diversity within ensembles can compensate for individual model limitations, yielding more accurate predictions of drug–ion channel interactions and their effect on ion channel function.

Together, these strategies produced models that captured key features of closed, open, and inactivated channel states. Our study goes beyond modeling distinct hERG channel conformations; it also provides extensive validation of their functional and pharmacological relevance through MD simulations and integrated docking analyses. The closed-state model exhibits a constricted pore and deactivated voltage-sensing domains, providing a structural basis for hERG channel deactivation. The open-state model displays a widened, conductive pore consistent with physiological ion flow during cardiac repolarization. The inactivated-state model features a non-conductive SF primarily characterized by SF residue V625 carbonyl flipping away from the conduction pathway. It also reveals notable rearrangements in the pore cavity that enhance drug binding, which is consistent with experimentally measured increases in drug affinity and arrhythmia risk (*Ficker et al., 1998*; *Perrin et al., 2008*; *Perrin et al., 2008*; *Yang et al., 2020*), yet previously lacking a defined structural explanation. Nonetheless, these alterations may not occur concurrently under physiological conditions and could represent only a subset of the broader conformational changes that accompany channel inactivation.

To demonstrate the broader feasibility of this approach, we applied it to another ion channel system, Na$_V$1.5, as illustrated in *Figure 1—figure supplement 1*. In this example, a deactivated VSD II from the cryo-EM structure of Na$_V$1.7 (PDB 6N4R) (*Xu et al., 2019*), which was trapped in a deactivated state by a bound toxin, was used as a structural template. This guided AlphaFold to generate an Na$_V$1.5 model in which all four voltage sensor domains (VSD I–IV) exhibit S4 helices in varying degrees of deactivation. Compared to the cryo-EM open-state Na$_V$1.5 structure (PDB 6LQA) (*Li et al., 2021a*), the predicted model displays a visibly narrower pore, representing a plausible closed state. This example underscores the versatility of our strategy in modeling alternative conformational states across diverse ion channels.

## Molecular basis linking hERG inactivation to enhanced drug-binding affinity

While previous studies have proposed potential molecular mechanisms for hERG C-type inactivation (*Lau et al., 2024*; *Li et al., 2021b*), they have primarily concentrated on conformational shifts leading to a non-conductive SF. Our study takes a step further and sheds light on how these alterations extend to the pore region and subsequently impact drug binding as seen in experiments, which is an outstanding issue in safety pharmacology and drug development as hERG channel is a notorious drug anti-target (*Perrin et al., 2008*; *Sanguinetti and Tristani-Firouzi, 2006*; *Vandenberg et al., 2012*; *Yang et al., 2020*).

Our results suggest that specific structural rearrangements occurring during the open-to-inactivated transition of the hERG channel may fine-tune the geometry of the central cavity in ways that enhance ligand binding. In particular, the movement of the canonical drug-binding residue Y652, together with the reorientation of the S624 sidechain hydroxyl groups, slightly expands the hydrophobic pockets beneath the SF (*Figure 3a, b*, at $Z = 21$ Å), allowing certain drugs to bind more deeply. Additionally, a slight inward rotation of the S6 helix repositions F656 toward the pore axis, which likely enhances drug interactions through π-stacking and hydrophobic contacts and helps to reduce the likelihood of drug dissociation from the cavity. These changes may not occur simultaneously or uniformly and may vary in magnitude, as suggested by the range of conformations observed in the AlphaFold models. Importantly, this local remodeling likely represents just one aspect of the broader inactivation process predicted by AlphaFold and validated in our study, contributing to drug stabilization without necessarily capturing the full inactivated state.

Together, these structural changes may account for the increased binding affinity observed for some compounds in the hERG channel inactivated state (*Figure 6—figure supplement 1*), offering a structural mechanism that links this enhanced binding to elevated arrhythmogenic potential (*Ficker et al., 1998*; *Jaiswal and Goldbarg, 2014*; *Perrin et al., 2008*; *Yang et al., 2020*). It is important to note that drug interaction with other cardiac ion channels may modulate or offset this risk in vivo, and the net proarrhythmic liability is determined by a more complex interplay of multi-channel effects (*Colatsky et al., 2016*). Nevertheless, given the exceptional sensitivity of the hERG channel to a wide range of compounds, understanding its state-dependent binding mechanisms remains critical for predicting and mitigating cardiac safety risks during drug development.

## Multi-state drug docking with AlphaFold-derived conformations outperforms reliance on a single-state experimental structure in predicting drug potency

An additional novelty of our study was the development of a new computational modeling and simulation approach that allowed us to use the predicted affinities of drug binding from docking simulations and compare them directly to measured hERG channel inhibition potencies from electrophysiology experiments. We employed a simulated hERG functional model comprising five functional states that we have extensively validated in our earlier studies (*Romero et al., 2015*) and performed in silico electrophysiological experiments under the same conditions as reported in the experimental papers. In doing so, we could compute the relative hERG channel state probabilities during the experimental protocol. The channel state probabilities were then used to scale the computed affinities in each state from the docking simulations, allowing comparison of the overall computed relative affinity with experimentally reported relative potencies. Such an analysis created an opportunity for an 'apples-to-apples' comparison between structurally derived affinity predictions and abundant functional

measurements that have been conducted for half a century. This novel linkage can be readily expanded to other protein targets and any variety of drugs. The intersection of structural modeling, molecular docking, functional simulations, and supporting experimental data offers a comprehensive approach to understanding protein structures and their links to biological functions.

## Comparative mechanisms of inactivation in hERG and other K⁺ channels

For other K⁺ channels, dilation in the SF has been proposed to be potential C-type inactivation mechanisms (*Chandy et al., 2023*; *Cuello et al., 2010*; *Selvakumar et al., 2022*; *Tan et al., 2022*; *Tyagi et al., 2022*). While inactivation processes across various K⁺ channels may share some similarities, the associated conformational changes can adopt distinct differences due to small variations in the SF sequences, which could explain the observed variability in inactivation rates among hERG and other ion channels (*Vandenberg et al., 2012*). As an example, a study on Shaker-family channels suggested that a two-step widening process in the upper SF could be a mechanism for C-type inactivation (*Wu et al., 2025*). The SF of the Shaker-family channel has the sequence 'TVGYGD'. The first step involves a partial twist of the P-loop backbone, particularly involving the upmost SF residue D30' (D379 in $K_V1.2$), which originally stabilizes itself by interacting with W17' (W336 in $K_V1.2$). The second step is the reorientation of the upper-middle SF residue Y28' (Y377 in $K_V1.2$) upward, which normally participates in hydrogen bonds with nearby pore-helix residues, to fill some of the original volume occupied by D30'.

In contrast, the hERG SF is characterized by the sequence 'SVGFGN'. In the upper SF of the hERG channel, phenylalanine (F) replaces the tyrosine (Y) found in many K⁺ channels including Shaker and $K_V1.2$. Phenylalanine has a non-polar benzyl side chain, whereas tyrosine has a polar hydroxyl group attached to its benzene ring. The hydroxyl group in tyrosine can form additional hydrogen bonds, which may stabilize different SF conformations in other K⁺ channels compared to hERG. Similarly, at the outermost end of the SF of hERG, asparagine (N) replaces the aspartate (D) found in many other K⁺ channels. Asparagine is uncharged, while aspartate introduces a negative charge through its carboxylate group, which could explain the differences in ion coordination and gating dynamics. These structural differences may explain why the hERG channel adopts a similar but distinct SF rearrangement compared to other K⁺ channels (such as Shaker and $K_V1.2$) and can have a slightly different structural mechanism of the C-type inactivation.

## Limitations, opportunities, and broader implications

Despite the promising results, our study is not without limitations. While AlphaFold has demonstrated remarkable accuracy in numerous instances (*AlQuraishi, 2019*; *Jumper et al., 2021b*; *Varadi and Velankar, 2023*), it is important to note that the predicted models may not always be reliably accurate to assess drug binding (*Karelina et al., 2023*). Moreover, hERG channel inactivation and closure might encompass a range of states, as was shown for other ion channels (*Catterall et al., 2020*; *Goldschen-Ohm et al., 2013*; *Hite and MacKinnon, 2017*; *Li et al., 2024*; *Yao et al., 2024*), and the conformations we have identified could potentially represent just a few possibilities within this broad spectrum. Our models excluded the N-terminal PAS domain due to GPU memory limitations, despite its inclusion in initial templates. This omission may overlook its potential roles in gating kinetics and allosteric effects on drug binding (*Abi-Gerges et al., 2011*; *Goversen et al., 2019*; *Gustina and Trudeau, 2013*; *Harchi et al., 2018*; *Perissinotti et al., 2018*). Future research will explore the full-length hERG channel with enhanced computational resources to assess these regions' contributions to conformational state transitions and pharmacology.

As noted in recent studies, pLDDT scores are not reliable indicators for selecting alternative conformations (*Bryant and Noé, 2024*; *Chakravarty et al., 2024*). To address this, we performed a protein backbone dihedral angle analysis in the regions of interest to ensure that our evaluation captured a representative range of sampled conformations. GALigandDock docking results, while insightful, are provisional (*Maly et al., 2022*) and limited by a rigid protein backbone assumption, thus preventing observation of drug-binding-induced allosteric modifications (*Harris et al., 2024*). As such, the results presented here should be interpreted as qualitative indicators of state-dependent binding trends rather than definitive quantitative predictions. To achieve more accurate binding affinity estimates, future studies could leverage MD simulations, incorporating methods like Molecular Mechanics/Poisson–Boltzmann Surface Area to assess relative ligand-binding energies during MD trajectories

(*Miller et al., 2012*; *Ngo et al., 2024*; *Wang et al., 2016*). These insights could be further extended by integrating MD results with multiscale functional modeling approaches, as demonstrated in our earlier work (*DeMarco et al., 2021*; *Yang et al., 2020*).

Our approach currently relies on well-characterized systems with ample static structures, MD simulation data, and mutagenesis insights, as demonstrated with the hERG channel, which may limit its applicability to less-studied proteins. Recently, AlphaFold3 was released, incorporating a diffusion model that enables the prediction of proteins in complex with other proteins, small molecules, nucleic acids, and ions (*Abramson et al., 2024*). We plan to explore the applicability of our template-guided methodology in a follow-up study, leveraging AlphaFold3's advanced diffusion-based architecture to enhance protein conformational state predictions and state-specific drug docking, particularly given its improved capabilities for modeling small molecule–protein interactions.

Correlating simulated drug-binding affinities with experimental results remains inherently challenging. As demonstrated in multiple studies, drug-binding potency is highly dependent on the measurement technique used, resulting in different $IC_{50}$ values being reported for the same channel–drug pairing (*Alexandrou et al., 2006*; *Asai et al., 2021*; *Cameron et al., 2021*; *Chiu et al., 2004*; *Ficker et al., 1998*; *Hansen et al., 2006*; *Johnson and Trudeau, 2023*; *Kushida et al., 2002*; *Orvos et al., 2019*; *Paul et al., 2002*; *Perrin et al., 2008*; *Rampe et al., 1997*; *Suessbrich et al., 1997*; *Tanaka et al., 2014*; *Walker et al., 1999*; *Zhang et al., 1999*; *Zhou et al., 1998*) as was explored in detail in our recent study (*Gomis-Tena et al., 2020*). Additionally, knowing the binding free energies of a drug is not the complete story; binding rates such as $k_{on}$ (association rate) and $k_{off}$ (dissociation rate) are also crucial for a quantitative evaluation of drug binding to the channel.

In conclusion, this study advances our understanding of hERG channel structural dynamics and state-dependent drug binding, while also demonstrating the broader potential of AlphaFold2-based modeling workflows. Our findings provide a foundation for integrating deep learning-based structure prediction with simulation and functional modeling to study other ion channels and membrane proteins. As computational methods continue to evolve, including alternatives like RoseTTAFold (*Baek et al., 2021*) and ESMFold (*Lin et al., 2023*), such integrated approaches will be increasingly valuable for addressing complex questions in ion channel physiology and pharmacology, with important implications for cardiac drug safety and therapeutic development.

## Materials and methods
### Introduction to AlphaFold2
AlphaFold2 employs a deep learning architecture that integrates several innovative components, including residue pair representation comprising an architecture module that represents each possible pair of amino acid residues in the sequence. This pairwise representation captures the interactions between residues that determine protein folding. AlphaFold2 also applies an attention mechanism, which constitutes a transformer-based model (similar to the architecture used in natural language processing) to weigh the influence of different parts of the input data differently (*Jumper et al., 2021a*). In AlphaFold2, the effect is to emphasize interactions between certain amino acid residues more than others based on how they might impact folding. There is also a so-called Evoformer block within the learning model that specifically processes the evolutionary data from MSAs, enabling the model to effectively incorporate evolutionary information (*Jumper et al., 2021a*). After processing through the Evoformer, intermediate representations are used to predict the distances and angles between residues as part of an iterative feedback process. A critical feature of AlphaFold2 is its iterative refinement, where pairwise residue representations, MSAs, and initial structural predictions are recycled through the model multiple times, improving accuracy with each iteration.

Due to the time-intensive process of creating MSAs for AlphaFold2, the ColabFold (*Mirdita et al., 2022*) webserver was made to streamline protein structure prediction by combining MMseqs2 (*Mirdita et al., 2019*) sequence search toolkit with AlphaFold2, enhancing runtime efficiency while preserving high prediction accuracy. ColabFold is available at https://github.com/sokrypton/Colab-Fold (*Ovchinnikov and Steinegger, 2025*).

## hERG channel model generation with ColabFold

We modeled the hERG potassium channel in three functional states (closed, open, and inactivated) using ColabFold with tailored structural templates and configurations. The structural templates were assembled by first using ChimeraX (*Pettersen et al., 2021*) to superimpose all relevant PDB entries with the 'matchmaker' command. Unnecessary regions were then removed from the aligned models, leaving only the fragments intended for use as templates. These remaining segments were merged into a single model using the 'combine' command. This approach preserves the spatial arrangement of template regions as they are expected to appear in the AlphaFold prediction, which can be important for guiding the model toward a specific conformation. The following sections describe the template selection process for each conformational state.

### Closed-state template

For modeling the closed state, we used a structural fragment from an experimental structure of a homologous protein that exhibits the desired characteristics of the target state. Specifically, for modeling the closed state of the hERG channel, we require the voltage sensor to be in a deactivated conformation.

We selected the deactivated VSD from the closed-state rat EAG channel cryo-EM structure (PDB 8EP1, residues H208–H343) (*Mandala and MacKinnon, 2022*) as the template to guide AlphaFold2 toward predicting a deactivated VSD conformation.

We combined this with the SF and adjacent pore helix from the open-state hERG cryo-EM structure (PDB 5VA2, residues I607–T634) (*Wang and MacKinnon, 2017*) to maintain its conductive conformation, as it is generally understood that K$^+$ channel closure primarily involves the intracellular gate rather than significant SF distortion. Including additional helices (e.g., S5–S6) or the entire membrane domain from PDB 8EP1 risked biasing the model toward the EAG channel's pore structure, which differs from hERG's, while omitting the cytosolic domain ensured focus on the VSD-driven closure without over-constraining cytoplasmic domain interactions.

### Open-state template

For modeling the open state of the hERG channel, we utilized the existing cryo-EM structure of the hERG channel (PDB 5VA2) (*Wang and MacKinnon, 2017*). This structure was shown in our previous studies to be open and conducting (*Miranda et al., 2020*; *Yang et al., 2020*). We rebuilt the missing extracellular loops using the Rosetta LoopRemodel protocol (*Huang et al., 2011*; *Leman et al., 2020*) to generate a complete model that serves as the basis for MD and drug docking simulations.

However, we also wanted to test the potential for AlphaFold2 to emulate the open-state model. To do so, specific regions of the putative open state hERG model (PDB 5VA2) (*Wang and MacKinnon, 2017*), namely the VSD (W398–V549), the SF with adjacent pore helix (I607–T634), a part of S6 helix and the cytosolic domain (S660–R863), were provided to ColabFold as structural templates. Then, 100 diverse models were generated for further analysis.

### Inactivated-state template

In situations where structural information about state transitions is limited or inconsistent, we adopt a second strategy. We erase regions expected to undergo conformational changes during state transition from an existing protein structure. For hERG inactivation in particular, where we know the SF shifts from an open, conductive conformation to a distorted, non-conductive state (*Miranda et al., 2020*), we initially used only the cytosolic domain from the open-state PDB 5VA2 (residues S660–R863) as a template. Excluding the SF or attached helices at this stage avoided locking the model into the open-state SF, and the cytosolic domain alone provided a minimal scaffold to maintain hERG's intracellular architecture without dictating pore dynamics. Following the initial prediction, we initiated more extensive sampling by using one of the predicted SFs that differs from the traditional open-state SF (PDB 5VA2) (*Wang and MacKinnon, 2017*) as a structural seed, aiming to guide predictions away from the open-state configuration. The VSD and cytosolic domain were also included in this state to discourage pore closure during prediction.

## ColabFold configuration

Structural templates were converted to CIF format and renamed '5va2' (after hERG cryo-EM structure) to meet ColabFold's four-letter code requirement. We optimized the following settings based on prior studies (*Brown et al., 2024*; *Del Alamo et al., 2022*; *Mirdita et al., 2022*; *Sala et al., 2023*) to sample diverse conformations:

- *max_msa* = '256:512': limit to 256 cluster centers and 512 sequences (down from 512:1024).
- *num_seeds* = 20: generate 5 models per seed, yielding 100 models per state, except for the initial inactivated-state phase (1 seed, 5 models).
- *use_dropout* = True: enable stochastic sampling for ambiguous regions.
- *num_recycles* = 20, *recycle_early_stop_tolerance* = 0.5: recycle up to 20 iterations, stop if pLDDT deviation fell below 0.5 after previous recycle.

It is important to note that the length of the structural template often affects the diversity of the predictions. Using a template that is too long or positioned differently may cause AlphaFold to generate models that do not reflect the template's features, and this behavior can vary depending on the system. AlphaFold's behavior is shaped by nonlinear patterns learned from vast structural data, making its internal logic not fully transparent. However, through careful tuning and testing, it is possible to influence its outputs by experimenting with input templates that vary in both length and position, as demonstrated in this study. As an example, *Figure 1—figure supplement 1* demonstrates how this approach can be applied to model the closed, resting state of Na$_V$1.5 using a structural template derived from the Na$_V$1.7 channel.

For our study, the N-terminal PAS domain (residues M1–R397) was not included in the final prediction due to graphics card memory limitation making the resultant model (W398–R863) resemble hERG 1b isoform (*Phartiyal et al., 2007*).

## Clustering of predicted models

The resulting 100 models for each structural state were categorized into clusters based on all-atom RMSD between those models. Closed-state models were clustered with a threshold of 0.75 Å across the entire channel, whereas inactivated- and open-state models focused on the SF (residues S624–G628), with a more stringent threshold of 0.35 Å.

We ranked the models in the cluster by their average per-residue confidence metric (pLDDT), which assesses the likelihood that the predicted structure aligns with an experimentally determined structure (*Jumper et al., 2021a*). pLDDT value above 90 is considered to be highly reliable, and those between 70 and 90 as reliable with generally good protein backbone structure prediction (*Jumper et al., 2021a*). Lower scores indicate regions of lower confidence and may be unstructured. Cluster 1 is defined in this study to be the cluster with the highest average pLDDT among all the clusters.

## Structural refinement

Afterward, we refined the preliminary atomistic structural models putatively representing each functional state of the hERG channel (open, inactivated, and closed) using the Rosetta FastRelax protocol (*Fleishman et al., 2011*; *Leman et al., 2020*) with an implicit membrane to optimize each residue conformation and resolve any steric clashes. The protocol was set to repeat 15 times and included an implicit POPC membrane environment. For each final model, 10 separate relaxation runs were executed, and the highest scoring model from these runs was selected for further simulations and analyses.

## Atomistic MD simulations to evaluate hERG channel conduction

### System assembly

The CHARMM-GUI web server (*Jo et al., 2008*) was employed to embed hERG channel structural models within tetragonal patches of phospholipid bilayers, each comprising approximately 230–240 POPC lipid molecules. The resulting assemblies were immersed in a 0.3 M KCl aqueous solution, yielding molecular systems with an approximate total of 138,000–144,000 atoms. Residue protonation reflected a pH of 7.4, with subunits terminated with standard charged N- and C-termini.

## Simulation setup

MD simulations were conducted using the Amber22 (*Case et al., 2005*) software suite. The simulations utilized standard all-atom Chemistry at Harvard Macromolecular Mechanics (CHARMM) force fields such as CHARMM36m (*Huang et al., 2017*) for proteins, C36 (*Klauda et al., 2010*) for lipids, and standard ion parameters (*Beglov and Roux, 1994*), in conjunction with the TIP3P water model (*Jorgensen et al., 1983*).

The systems were maintained at 310.15 K and 1 atm pressure in the isobaric–isothermal (*NPT*) ensemble, facilitated by Langevin thermostat and the Berendsen barostat. Non-bonded interactions were calculated with a cutoff of 12 Å. Long-range electrostatic forces were computed using the Particle Mesh Ewald method (*Darden et al., 1993*), and van der Waals interactions were not subjected to long-range correction as per recommendations for the C36 lipid force field (*Klauda et al., 2010*). All hydrogen-connected covalent bonds were constrained using the SHAKE algorithm to enable a 2-fs MD simulation time step (*Ryckaert et al., 1977*).

## MD equilibration protocol and production run

Equilibration began with harmonic restraints imposed on all protein atoms and lipid tail dihedral angles, initially set at 20 kcal/mol/Å² and reduced to 2.5 kcal/mol/Å² over ~2 ns. A subsequent 90 ns equilibration phase further decreased the restraints to 0.1 kcal/mol/Å², initially encompassing all atoms in the protein and eventually focusing solely on the backbone atoms of pore-domain residues (G546 to F720). In selected MD simulations, an electric field was applied along the *Z* direction to mimic membrane voltage (*Gumbart et al., 2012*), increasing linearly over the final 10 ns of equilibration to reach either 500 or 750 mV. This setup prefaced a production phase lasting 910 ns, totaling 1 μs of total simulation time per each case.

## Docking of small-molecule drugs to hERG channel models

### Ligand preparation

Ligand structures (i.e., drugs) were retrieved from the PubChem (*Kim et al., 2016*) and ZINC (*Tingle et al., 2023*) databases. In this study, we considered the protonation states of the top two most dominant species at the physiological pH of 7.4, computed using the Henderson–Hasselbalch equation. After these initial modifications, each ligand's partial atomic charges, as well as atom and bond types, were refined using AM1-BCC correction via the Antechamber module in AmberTools22 (*Case et al., 2023*; *Case et al., 2005*).

Prior to the docking process, each ligand was individually positioned within the pore of the hERG channel, specifically between the key drug-binding residues, Y652 and F656, located on the pore-lining S6 helices using ChimeraX (*Pettersen et al., 2021*).

### Drug docking

Docking was executed using the GALigandDock (*Park et al., 2021*) Rosetta mover, a component of the Rosetta software suite (*Leman et al., 2020*). The *DockFlex* mode was utilized for this purpose with a spatial padding of 6 Å. For every individual hERG channel–ligand pair, a substantial collection of 25,000 docking poses was generated for each ligand–hERG channel complex. This extensive array of poses was intended to ensure a comprehensive exploration of potential ligand-binding configurations and orientations within the hERG channel pore.

### Clustering the results

The top 100 lowest energy poses were selected and clustered based on the structural similarity of ligand positions, using all-atom RMSD while accounting for the fourfold symmetry of the hERG channel. To identify the most representative binding mode, we applied a hybrid scoring approach that considers both binding energy and cluster size. Clusters with similar average binding energies (within a defined tolerance of ± 0.25 kcal/mol from the best-scoring cluster) were compared, and preference was given to non-outlier clusters with larger numbers of poses. The cluster with the most favorable balance of energy and convergence was considered the best and selected for further analysis, and its mean binding energy and standard deviation were used to represent the drug–channel interaction. This approach was designed to reduce sensitivity to isolated low-energy poses that may not reflect

**Table 5.** Transition rates in the hERG channel ($I_{Kr}$) Markov model.

| | Transition rates (ms$^{-1}$) Drug free Kr channel |
|---|---|
| C3→C2 | $ae = \frac{T}{T_{base}} e^{\left(24.335 + \frac{T_{base}}{T}\left(0.0112 \times V - 25.914\right)\right)}$ |
| C2→C3 | $be = \frac{T}{T_{base}} e^{\left(13.688 + \frac{T_{base}}{T}\left(-0.0603 \times V - 15.707\right)\right)}$ |
| C2→C1 | $ain = \frac{T}{T_{base}} e^{\left(22.746 + \frac{T_{base}}{T}\left(-25.914\right)\right)}$ |
| C1→C2 | $bin = \frac{T}{T_{base}} e^{\left(13.193 + \frac{T_{base}}{T}\left(-15.707\right)\right)}$ |
| C1→O | $aa = \frac{T}{T_{base}} e^{\left(22.098 + \frac{T_{base}}{T}\left(0.0365 \times V - 25.914\right)\right)}$ |
| O→C1 | $bb = \frac{T}{T_{base}} e^{\left(7.313 + \frac{T_{base}}{T}\left(-0.0399 \times V - 15.707\right)\right)}$ |
| O→I | $\beta i = \frac{T}{T_{base}} e^{\left(30.016 + \frac{T_{base}}{T}\left(0.0223 \times V - 30.88\right)\right)} \times \left(\frac{5.4}{[K]^{o}}\right)^{0.4}$ |
| I→O | $\alpha i = \frac{T}{T_{base}} e^{\left(30.061 + \frac{T_{base}}{T}\left(-0.0312 \times V - 33.243\right)\right)}$ |
| Base temperature ($T_{base}$) | 310 K |
| Temperature ($T$) | Please see **Table 6** |

stable binding modes and to favor interactions that are both energetically favorable and structurally well-converged.

## Visualization

Within the selected cluster, we chose the top scoring pose as the representative pose for further analysis. This pose was then subjected to a detailed analysis of protein–ligand interactions utilizing the Grapheme Toolkit from the OpenEye software suite (https://www.eyesopen.com/). The criteria for detecting interactions are outlined in the OEChem Toolkit manual (https://docs.eyesopen.com/tool-kits/python/oechemtk/OEBioClasses/OEPerceiveInteractionOptions.html), with two modifications to minimize clutter: the *MaxContactFraction* is set to 1 (default: 1.2), and the *MaxCationPiAngle* is adjusted to 30° (default: 40°). The interaction patterns and binding sites were subsequently rendered as a two-dimensional image for comprehensive visual interpretation. Additionally, for a more detailed understanding of the spatial arrangement, three-dimensional visualization of the protein-ligand complexes was conducted using the ChimeraX (*Pettersen et al., 2021*) software.

## Comparing simulated and experimental drug-binding affinities

### Five-state hERG Markov model for state probability prediction

Over the time course of experimental recordings of hERG inhibition by various drugs, the channel can be in different functional states, each bound by drugs with different ionization states, making it difficult to compare experimental and simulated binding affinities. Moreover, different studies utilize different electrophysiological protocols to measure state-dependent ligand binding, further compli-cating comparisons.

To address this complication, we used a five-state hERG Markov model to predict the probabilities of the channel in each state (open, closed, and inactivated) during a given experimental protocol (*Romero et al., 2015*). Transition rate constants are provided in *Table 5*. The protocols that were used in the model to calculate each state probabilities (closed states: C3 + C2 + C1, inactivated state: I, and open state: O) are shown in *Table 6*. To simulate the inhibitory effects of the drug on the hERG channel current, $I_{Kr}$, we decreased the peak conductance, $G_{Kr}$, in a concentration-dependent fashion using a concentration response relationship with a Hill coefficient of 1 ($n = 1$) as follows:

$$G_{\text{Kr}} = G_{\text{Kr,max}} \cdot \left( \frac{1}{1 + \left( \frac{[\text{Drug}]}{\text{IC}_{50}} \right)^n} \right)$$

where $G_{\text{Kr,max}}$ is the nominal conductance value obtained from each ventricular myocyte model, [Drug] is a molar drug concentration, and the $\text{IC}_{50}$ is the concentration of drug that produces a 50% inhibition of the targeted transmembrane current, that is, $I_{\text{Kr}}$ in this case (see *Table 6*).

## Calculation of simulated drug-binding affinities

To obtain the simulated binding affinity of a drug, $\Delta G_{\text{bind, sim}}$, we multiplied the binding affinity for each state by the probability of the channel being in that state and combined these values for both neutral and cationic forms of the drug, as represented by the following equation:

$$\begin{aligned}
\Delta G_{\text{bind, sim}} = P_{\text{drug, neutral}} \big( &\Delta G_{\text{bind, O, neutral}} \times P_{\text{hERG, O}} + \Delta G_{\text{bind, I, neutral}} \times P_{\text{hERG, I}} \\
&+ \Delta G_{\text{bind, C, neutral}} \big) + P_{\text{drug, cationic}} \big( \Delta G_{\text{bind, O, cationic}} \times P_{\text{hERG, O}} \\
&+ \Delta G_{\text{bind, I, cationic}} \times P_{\text{hERG, I}} + \Delta G_{\text{bind, C, cationic}} \times P_{\text{hERG, C}} \big)
\end{aligned}$$

Here,

- $\Delta G_{\text{bind, O}}$, $\Delta G_{\text{bind, I}}$, $\Delta G_{\text{bind, C}}$ represent simulated binding affinities for the open, inactivated, and closed state, respectively, to either the neutral or the cationic form of the drug.
- $P_{\text{hERG,O}}$, $P_{\text{hERG,I}}$, $P_{\text{hERG,C}}$ represent the fraction of channels that are in the open, inactivated, and closed state, at the time when the tail current was observed in electrophysiological recordings to calculate drug fractional block, determined for the specific voltage protocol employed.
- $P_{\text{drug,neutral}}$ and $P_{\text{drug, cationic}}$ represent the fraction of neutral and cationic species of each drug at physiological pH 7.4 as calculated using the Henderson–Hasselbalch equation using drug $pK_a$ values from Chemaxon. The zwitterionic species of moxifloxacin instead of cationic was included. These data are recorded in *Table 4*.

Experimental $\text{IC}_{50}$ values (in units of M) were converted to equivalent binding free energies using the equation $\Delta G_{pot, exp} = -RT\ln(1/\text{IC}_{50})$ where $R = 0.0019872036$ kcal K$^{-1}$ mol$^{-1}$ is the gas constant and $T$ is the experimental temperature in K.

## Molecular graphics and interaction analysis

Molecular graphics visualization was performed using ChimeraX (*Pettersen et al., 2021*). MD trajectory and simulation images were visualized using VMD (*Humphrey et al., 1996*). Interaction network analysis was performed using the Protein-Ligand Interaction Profiler (PLIP) (*Salentin et al., 2015*) with criteria outlined in *Table 1*.

## Acknowledgements

We would like to thank all members of the IV, CEC, and VY-Y laboratories and KN's cats, Momo and Orange, for helpful discussions and support. This work was supported by National Institutes of Health Common Fund Grant OT2OD026580 (to CEC and IV), National Heart, Lung, and Blood Institute (NHLBI) grants R01HL128537, R01HL174001, R01HL085844, R01HL152681, and U01HL126273 (to CEC, VY-Y, and IV), American Heart Association Career Development Award grant 19CDA34770101 (to IV), National Science Foundation travel grant 2032486 (to IV), UC Davis Department of Physiology and Membrane Biology Research Partnership Fund (to CEC and IV) as well as UC Davis T32 Predoctoral Training in Basic and Translational Cardiovascular Medicine fellowship supported in part by NHLBI Institutional Training Grant T32HL086350 (to KN). Computer allocations were provided through Advanced Cyberinfrastructure Coordination Ecosystem: Services & Support (ACCESS) grant MCB170095 (to IV, CEC, and VY-Y), Texas Advanced Computing Center (TACC) Leadership Resource and Pathways Allocations MCB20010 (IV, CEC, and VY-Y), Oracle for Research fellowship and cloud credits award (to IV, CEC), Pittsburgh Supercomputing Center (PSC) Anton 2 allocations PSCA17085P, MCB160089P, PSCA18077P, PSCA17085P, and PSCA16108P (to IV, CEC, and VY-Y). Anton 2 computer time was provided by the Pittsburgh Supercomputing Center (PSC)

**Table 6.** Voltage stimulation protocols and $IC_{50}$ for drugs used in the $I_{kr}$ Markov model.

| Drugs | IC$_{50}$ (nM) | Dose (nM) | Temperature (K) | Voltage protocols | Refs |
|---|---|---|---|---|---|
| Astemizole | 26 | 80 | 310 |  | *Chiu et al., 2004* |
| | 31 | 100 | 310 |  | *Orvos et al., 2019* |
| Terfenadine | 30.6 | 30 | 310 |  | *Tanaka et al., 2014* |
| | 44.5 | 100 | 295 |  | *Rampe et al., 1997* |
| Cisapride | 6.9 | 20 | 310 |  | *Chiu et al., 2004* |
| | 143 | 500 | 295 |  | *Zhang et al., 1999* |
| Verapamil | 180.4 | 300 | 310 |  | *Johnson and Trudeau, 2023* |
| dl-Sotalol | 515,500 | 300,000 | 310 |  | *Perrin et al., 2008* |

*Table 6 continued on next page*

*Table 6 continued*

| Drugs | IC$_{50}$ (nM) | Dose (nM) | Temperature (K) | Voltage protocols | Refs |
|---|---|---|---|---|---|
| | 320 | 1000 | 295 | | *Ficker et al., 1998* |
| Dofetilide | 17.9 | 10 | 295 | | *Li et al., 2012* |
| Haloperidol | 1000 | 3000 | 295 | | *Suessbrich et al., 1997* |
| Amiodarone | 45 | 100 | 310 | | *Zhang et al., 2016* |
| E-4031 | 7.7 | 10 | 310 | | *Zhou et al., 1998* |
| | 28,300 | 20,000 | 295 | | |
| Clozapine | 22,900 | 20,000 | 295 | | *Lee et al., 2006* |
| NS1643 | 10,500 | 30,000 | 310 | | *Hansen et al., 2006* |

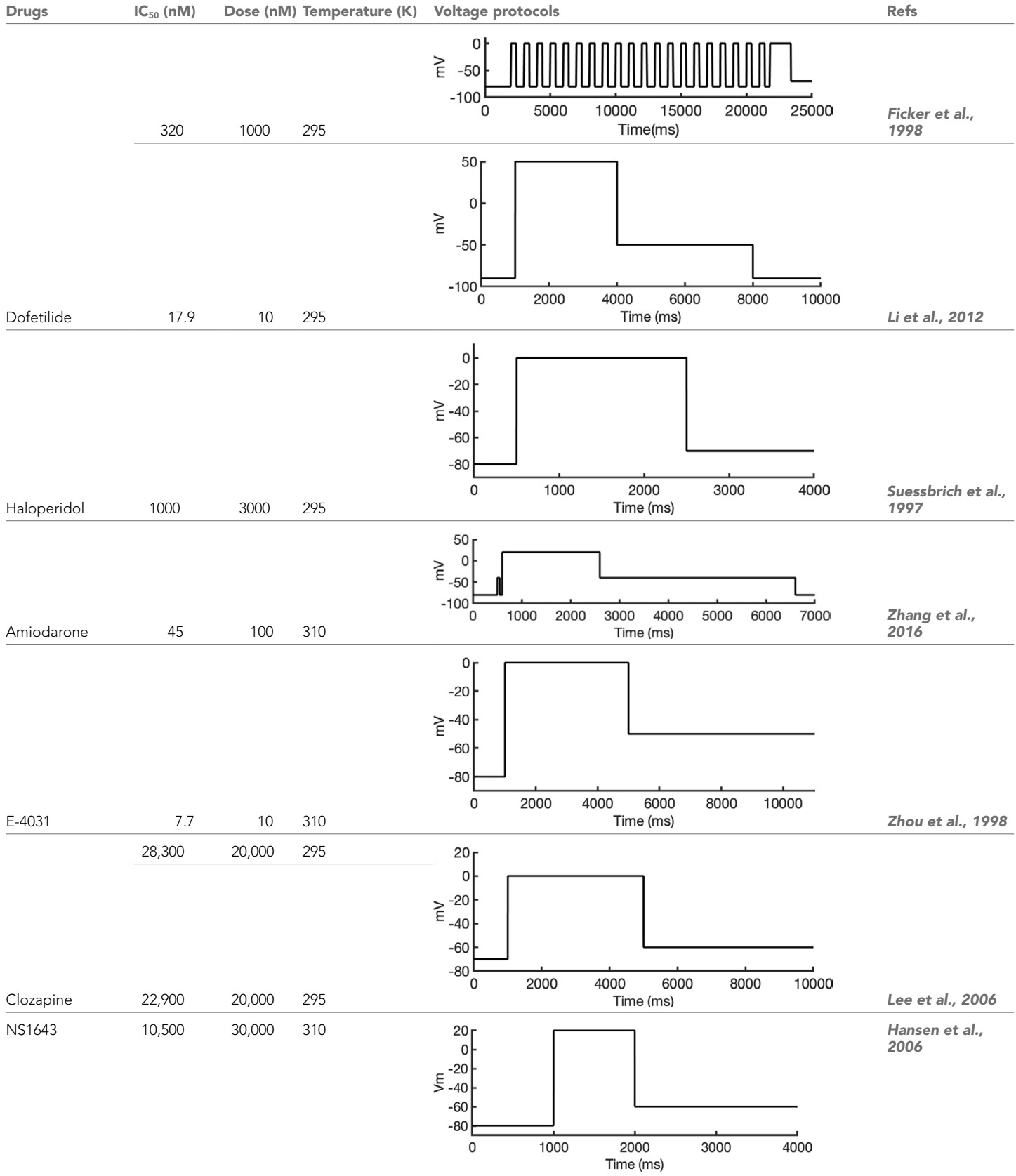

*Table 6 continued on next page*

*Table 6 continued*

| Drugs | IC$_{50}$ (nM) | Dose (nM) | Temperature (K) | Voltage protocols | Refs |
|---|---|---|---|---|---|
| | 65,000 | 60,000 | 295 | | |
| Moxifloxacin | 35,700 | 600,000 | 310 | | *Alexandrou et al., 2006* |
| | 800 | 500 | 295 | | *Yan et al., 2016* |
| Quinidine | 410 | 500 | 310 | | *Paul et al., 2002* |
| Perhexiline | 7800 | 1000 | 295 | | *Walker et al., 1999* |
| Nifekalant | 7900 | 1000 | 295 | | *Kushida et al., 2002* |

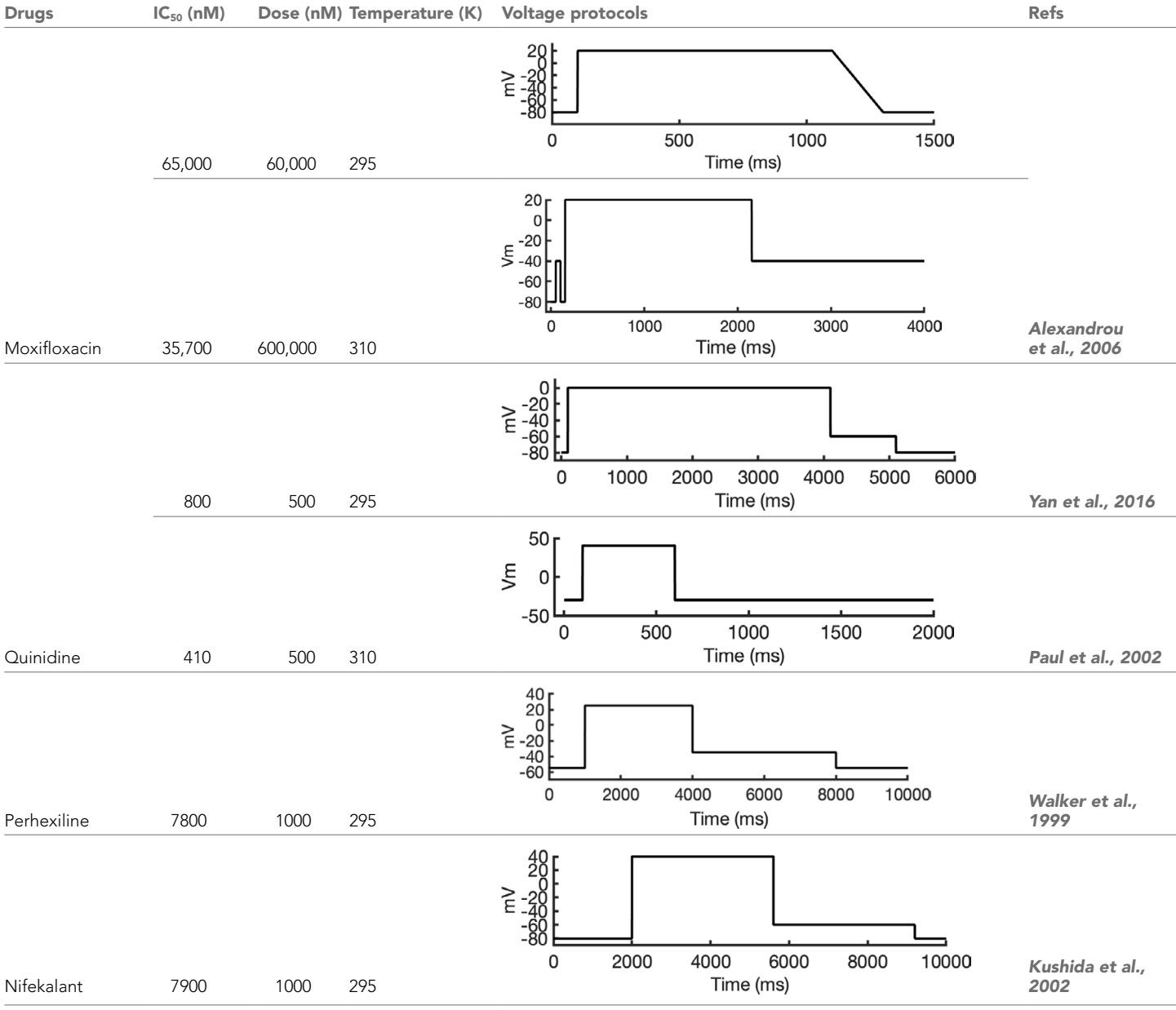

through Grant R01GM116961 from the National Institutes of Health. The Anton 2 machine at PSC was generously made available by DE. Shaw Research. OpenEye academic license was provided by OpenEye Scientific.

## Additional information

### Funding

| Funder | Grant reference number | Author |
|---|---|---|
| National Institutes of Health | OT2OD026580 | Colleen E Clancy Igor Vorobyov |

| Funder | Grant reference number | Author |
|---|---|---|
| National Heart, Lung, and Blood Institute | R01HL128537 | Vladimir Yarov-Yarovoy<br>Colleen E Clancy<br>Igor Vorobyov |
| National Heart, Lung, and Blood Institute | R01HL174001 | Vladimir Yarov-Yarovoy<br>Colleen E Clancy<br>Igor Vorobyov |
| National Heart, Lung, and Blood Institute | R01HL085844 | Vladimir Yarov-Yarovoy<br>Igor Vorobyov |
| National Heart, Lung, and Blood Institute | R01HL152681 | Colleen E Clancy<br>Igor Vorobyov |
| National Heart, Lung, and Blood Institute | U01HL126273 | Vladimir Yarov-Yarovoy<br>Colleen E Clancy<br>Igor Vorobyov |
| American Heart Association | 19CDA34770101 | Igor Vorobyov |
| National Science Foundation | 2032486 | Igor Vorobyov |
| National Heart, Lung, and Blood Institute | T32HL086350 | Khoa Ngo |
| University of California Davis School of Medicine | Department of Physiology and Membrane Biology Research Partnership Fund | Colleen E Clancy<br>Igor Vorobyov |
| Oracle | Oracle for Research fellowship | Igor Vorobyov |
| Advanced Cyberinfrastructure Coordination Ecosystem: Services & Support (ACCESS) | MCB170095 | Igor Vorobyov<br>Colleen E Clancy<br>Vladimir Yarov-Yarovoy |
| Texas Advanced Computing Center (TACC) | MCB20010 | Igor Vorobyov<br>Colleen E Clancy<br>Vladimir Yarov-Yarovoy |
| Pittsburgh Supercomputing Center (PSC) | PSCA17085P | Igor Vorobyov<br>Colleen E Clancy<br>Vladimir Yarov-Yarovoy |
| Pittsburgh Supercomputing Center (PSC) | MCB160089P | Igor Vorobyov<br>Colleen E Clancy<br>Vladimir Yarov-Yarovoy |
| Pittsburgh Supercomputing Center (PSC) | PSCA18077P | Igor Vorobyov<br>Colleen E Clancy |
| Pittsburgh Supercomputing Center (PSC) | PSCA16108P | Igor Vorobyov<br>Colleen E Clancy |

The funders had no role in study design, data collection, and interpretation, or the decision to submit the work for publication.

## Author contributions

Khoa Ngo, Conceptualization, Resources, Data curation, Software, Formal analysis, Validation, Investigation, Visualization, Methodology, Writing – original draft, Project administration, Writing – review and editing; Pei-Chi Yang, Resources, Data curation, Software, Formal analysis, Validation, Investigation, Methodology, Writing – original draft, Writing – review and editing; Vladimir Yarov-Yarovoy, Resources, Software, Supervision, Funding acquisition, Writing – original draft, Project administration, Writing – review and editing; Colleen E Clancy, Conceptualization, Resources, Supervision, Funding acquisition, Investigation, Methodology, Writing – original draft, Project administration, Writing – review and editing; Igor Vorobyov, Conceptualization, Resources, Software, Supervision, Funding

acquisition, Investigation, Methodology, Writing – original draft, Project administration, Writing – review and editing

## Author ORCIDs
Khoa Ngo https://orcid.org/0000-0002-7454-2924
Pei-Chi Yang https://orcid.org/0000-0002-5753-1131
Vladimir Yarov-Yarovoy https://orcid.org/0000-0002-2325-4834
Colleen E Clancy https://orcid.org/0000-0001-6849-4885
Igor Vorobyov https://orcid.org/0000-0002-4767-5297

Reviewer #1 (Public review): https://doi.org/10.7554/eLife.104901.3.sa1
Reviewer #2 (Public review): https://doi.org/10.7554/eLife.104901.3.sa2
Author response https://doi.org/10.7554/eLife.104901.3.sa3

# Additional files

### Supplementary files
Supplementary file 1. Open-state hERG channel structural model (PDB 5VA2-based).

Supplementary file 2. Inactivated-state AlphaFold hERG channel structural model.

Supplementary file 3. Closed-state AlphaFold hERG channel structural model.

Supplementary file 4. Open-state AlphaFold hERG channel structural model (ic3).

MDAR checklist

### Data availability
All final study data are included in the article and/or supplementary files, with key molecular modeling, docking, molecular dynamics simulation, analysis data files, and scripts available to download from Dryad Digital Repository. Scripts developed in this study for analyzing AlphaFold-predicted protein structure models are also available on GitHub (copy archived at *Ngo, 2025*).

The following dataset was generated:

| Author(s) | Year | Dataset title | Dataset URL | Database and Identifier |
| --- | --- | --- | --- | --- |
| Ngo K, Yang P, Yarov-Yarovoy V, Clancy CE, Vorobyov I | 2025 | Data from: Harnessing AlphaFold to reveal hERG channel conformational state secrets | https://doi.org/10.5061/dryad.18931zd5x | Dryad Digital Repository, 10.5061/dryad.18931zd5x |

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
