## [Editor Report · eLife Assessment]

This **valuable** study uses AlphaFold2 to guide the structural modelling of different states of the human voltage-gated potassium channel K_V_11.1, a key pharmacological drug target. Follow-up molecular dynamics and drug-docking simulations, combined with experimental characterization, offer **convincing** evidence supporting the models. The work shows potential for improving drug potency predictions in ion channel pharmacology.

---

## [Referee Report · Reviewer #1 (Public review)]

Summary:

Ngo et. al use several computational methods to determine and characterize structures defining the three major states sampled by the human voltage-gated potassium channel hERG: the open, closed and inactivated state. Specifically, they use AlphaFold and Rosetta to generate conformations that likely represent key features of the open, closed and inactivated states of this channel. Molecular dynamics simulations confirm that ion conduction for structure models of the open but not the inactivated state. Moreover, drug docking in silico experiments show differential binding of drugs to the conformation of the three states; the inactivated one being preferentially bound by many of them. Docking results are then combined with a Markov model to get state-weighted binding free energies that are compared with experimentally measured ones.

Strengths:

The study uses state-of-the-art modeling methods to provide detailed insights into the structure-function relationship of an important human potassium channel. AlphaFold modeling, MD simulations and Markov modeling are nicely combined to investigate the impact of structural changes in the hERG channel on potassium conduction and drug binding.

Weaknesses:

(1) Selection of inactivated conformations based on AlphaFold modeling seems a bit biased.

The authors base their initial selection of the "most likely" inactivated conformation on the expected flipping of V625 and the constriction at G626 carbonyls. This follows a bit the "Streetlight effect". It would be better to have selection criteria that are independent of what they expect to find for the inactivated state conformations. Using cues that favour sampling/modeling of the inactivated conformation, such as the deactivated conformation of the VSD used in the modeling of the closed state, would be more convincing. There may be other conformations that are more accurately representing the inactivated state. In addition, I am not sure whether pLDDT is a good selection criterion. It reports on structural confidence, but that may not relate to functional relevance.

(2) The comparison of predicted and experimentally measured binding affinities lacks of appropriate controls. Using binding data from open-state conformations only is not the best control. A much better control is the use of alternative structures predicted by AlphaFold for each state (e.g. from the outlier clusters or not considered clusters) in the docking and energy calculations. Importantly, labels for open, closed and inactivated state should be randomized to check robustness of the findings. Such a control would strengthen the overall findings significantly.

(3) Figures where multiple datapoints are compared across states generally lack assessment of the statistical significance of observed trends (e,g. Figure 3d).

The authors have successfully achieved their goal of providing new insights into the structural details of the three major conformational states sampled by the human voltage-gated potassium channel hERG, and linking these states to changes in drug-binding affinities. However, the study would benefit from more robust controls and orthogonal validation. Additionally, the generalizability of the approach remains to be demonstrated.

---

## [Referee Report · Reviewer #2 (Public review)]

Summary:

Ngo et al. use AlphaFold2 and Rosetta to model closed, open, and inactive states of the human ion channel hERG. Subsequent MD simulations and comparisons with experiment support the plausibility of their models.

Strengths:

Ngo et al. employ various computational methods to enhance AlphaFold2's prediction capabilities for the human voltage-gated potassium channel hERG. They guide AlphaFold2 to explore different protein conformations and states, including its open, closed, and inactivated forms, using targeted templates. Additionally, they applied the Rosetta FastRelax protocol with an implicit membrane to refine the conformation of each residue in the predictions and address steric clashes, along with molecular dynamics (MD) simulations to account for membrane-pore flexibility. The methodology is well-described, and the figures are clear and descriptive.

The authors have addressed some of the concerns raised during the first round of reviews. For instance, to mitigate potential bias in selecting the inactivated conformation, they evaluated conformational variability via backbone dihedral angles at specific residues in the selectivity filter and the drug binding sites. They also evaluated the top representative model from inactivated-state-sampling Cluster 3 (termed "AF ic3"), which was initially excluded. This model is now included in the revised manuscript as Figure S9a, b. MD simulations confirmed that this state could be a potential alternative open-state conformation. The authors also acknowledged the limitation of their study by not incorporating other enhanced sampling methods and AF3.

In the revised manuscript, the authors provided more extensive explanations of their methods. For example, they explained that their approach to template selection was guided by their experience-AlphaFold2 with larger templates often overly constraining predictions to the input structure, reducing its flexibility to explore alternative conformations. In contrast, smaller, targeted fragments increase the likelihood that AlphaFold2 will incorporate the desired structural features while predicting the rest of the protein. They also noted that pLDDT scores are not always reliable for selecting new or alternative conformations, citing proper references. They included a model from cluster 3 of the inactivated-state sampling process, which exhibited lower pLDDT scores to illustrate this further.

Another point raised by the reviewers was the exclusion of the N-terminal PAS domain due to GPU memory limitations and its impact on the study. This omission may overlook the PAS domain's potential roles in gating kinetics and allosteric effects on drug binding. The authors acknowledged these limitations in the main text and highlighted the need for future studies to explore these regions in greater detail. They also alluded to potential future research to address these points. Additionally, they have made some of their analysis scripts and tools available on GitHub as a community resource.

Weakness:

The primary issue with the study is the lack of a general pipeline or strategy that can be universally applied to any system, even if limited to ion channels or membrane proteins. A related paper assessed the conformational variability in voltage-sensing domains (VSDs) by applying both the default MSA depth and a range of reduced MSA depths to enhance conformational diversity (please see https://doi.org/10.1101/2025.03.12.642934). They generated 600 models for 32 members of the voltage-gated cation channel superfamily and demonstrated that AlphaFold2 can predict a range of diverse structures of the VSDs, representing activated, deactivated, and intermediate conformations, with more diversity observed for some VSDs compared to others.

The authors have addressed one of the reviewer's concerns about generalizability by including an example in Figure S14 of the modified text, showing how their approach can be applied to model another ion channel system. However, some outstanding questions remain: Is this method better suited for ion channels or membrane proteins with already solved structures and extensive research available? Can this pipeline be applied to other systems as well? Additionally, how does this method compare to other methods using MSA subsampling and other enhanced AF-based techniques to generate alternative conformations of proteins?

---

## [Author Response]

The following is the authors’ response to the original reviews

**Reviewer #1 (Public review):**
Weaknesses:(1) The selection of inactivated conformations based on AlphaFold modeling seems a bit biased. The authors base their selection of the “most likely” inactivated conformation on the expected flipping of V625 and the constriction at G626 carbonyls. This follows a bit of the “Streetlight effect”. It would be better to have selection criteria that are independent of what they expect to find for the inactivated state conformations. Using cues that favour sampling/modeling of the inactivated conformation, such as the deactivated conformation of the VSD used in the modeling of the closed state, would be more convincing. There may be other conformations that are more accurately representing the inactivated state. I see no objective criteria that justify the non-consideration of conformations from cluster 3 of the inactivated state modeling. I am not sure whether pLDDT is a good selection criterion. It reports on structural confidence, but that may not relate to functional relevance.

We sincerely thank the reviewer for their perceptive critique highlighting potential bias in selecting the inactivated conformation. We recognize that over-relying on preconceived traits could limit exploration of diverse inactivated states, and we appreciate the opportunity to address this concern.

Although we selected the model with the flipped V625 in the selectivity filter (SF) from the first round of inactivated-state sampling as the template for the second round, the resulting models still exhibited substantial diversity in their SF conformations. This selection primarily served to steer predictions away from the open-state configuration observed in the PDB 5VA2 SF, and we have clarified this rationale in the Methodology section. To assess conformational variability, we examined backbone dihedral angles (phi φ and psi ψ) at key residues in the selectivity filter (S624 – G628) and drugbinding region on the pore-lining S6 segment (Y652, F656), of all 100 models sampled in the subsequent inactivatedstate-sampling attempt. By overlaying the φ and ψ dihedral angles from different models, including the open state (PDB 5VA2-based), the closed state, and representative models from AlphaFold inactivated-state-sampling Cluster 2 and Cluster 3, we found that these conformations consistently fall within or near high-probability regions of the dihedral angle distributions. This indicates that these structural states are well represented within the ensemble of conformations sampled by AlphaFold within the scope of this study, particularly at functionally critical positions.

Following the analysis above and consistent with the reviewer’s suggestion, we evaluated the top representative model from inactivated-state-sampling Cluster 3 (named “AF ic3”), which we had initially excluded. This model demonstrated SF residue G626 carbonyl oxygen flipped away from the conduction pathway, hinting at potential impact on ion conduction, yet its pore region structurally resembled the open state (Figure S9a, b). To test this objectively, we ran molecular dynamics (MD) simulations (2 runs, 1 μs long each, with applied 750 mV voltage) with varied initial ion/water configurations in the SF, finding it consistently open and conducting throughout (Figure S9c, d), consistent with our previous observations in Figure S11 that ion conduction can still occur when the upper SF is dilated. Drug docking (Figure S12) further revealed that the model exhibited binding affinities similar to those for the PDB 5VA2-based openstate structure. These findings combined led us to classify it as a possible alternative open-state conformation.

Models from Cluster 4 were not tested due to extensive steric clashes, where residues in the SF overlapped with neighboring residues from adjacent subunits. The remaining models displayed SF conformations that combined features from earlier clusters. However, due to subunit-to-subunit variability, where individual subunits adopted differing conformations, they were classified as outliers. This combination of features may be valuable to investigate further in a follow-up study.

We acknowledge that our approach is just one of many ways to sample different states, and alternative strategies, such as generating more models, varying multiple sequence alignment (MSA) subsampling, or testing different templates, might reveal improved models. Given that hERG channel inactivation likely spans a spectrum of conformations, our resource limitations may have restricted us to exploring and validating only part of this diversity. Nevertheless, the putative inactivated (AlphaFold Cluster 2) model’s non-conductivity and improved affinity for drugs targeting the inactivated state observed in our study suggests that this approach may be capturing relevant features of the inactivated-state conformation. We look forward to investigating deeper other possibilities in a future study and are grateful for the reviewer’s feedback.

(2) The comparison of predicted and experimentally measured binding affinities lacks an appropriate control. Using binding data from open-state conformations only is not the best control. A much better control is the use of alternative structures predicted by AlphaFold for each state (e.g. from the outlier clusters or not considered clusters) in the docking and energy calculations. Using these docking results in the calculations would reveal whether the initially selected conformations (e.g. from cluster 2 for the inactivated state) are truly doing a better job in predicting binding affinities. Such a control would strengthen the overall findings significantly.

We appreciate the reviewer’s insightful suggestion. To address this, we extended our analysis by incorporating an alternative AlphaFold2-predicted model from inactivated-state-sampling cluster 3 as a structural control. This model was established in a previously discussed analysis to be open and conducting as a follow up to comment #1, so we will call it Open (AF ic3) to differentiate it from Open (PDB 5VA2). We evaluated this new model in single-state and multi-state contexts alongside our original open-state model based on the experimental PDB 5VA2 structure. Additionally, we expanded the drug docking procedure to explore a broader region around the putative drug binding site by increasing the sampling space, and we adopted an improved approach for selecting representative docking poses to better capture relevant binding modes.

Shown in Figure 7 are comparisons of experimental drug potencies with the binding affinities from the molecular docking calculations under the following conditions:

(a) Single-state docking using the experimentally derived open-state structure (PDB 5VA2)

(b) Multi-state docking incorporating open (PDB 5VA2), inactivated, and closed-state conformations weighted by experimentally observed state distributions

(c) Single-state docking using an alternative AlphaFold-predicted open-state (inactivated-state-sampling cluster 3, AF ic3)

(d) Multi-state docking combining the AlphaFold-predicted open-state (inactivated-state-sampling cluster 3, AF ic3)

Using only the open-state model (PDB 5VA2) yielded a moderate correlation with experimental data (*R2* = 0.43, *r* = 0.66, Figure 7a). Incorporating multi-state binding (weighted by their experimental distributions) improved the correlation substantially (*R2* = 0.63, *r* = 0.79, Figure 7b), boosting predictive power by 47% and underscoring the value of multi-state modeling. Importantly, this improvement was achieved without considering potential drug-induced allosteric effects on the hERG channel conformation and gating, which will be addressed in future work.

Next, we substituted the PDB 5VA2-based open-state model with the AF ic3 open-state model. Docking to this alternative model alone produced similar performance (*R2* = 0.44, *r* = 0.66, Figure 7c), and incorporating it into the multi-state ensemble further improved the correlation with experiments (*R2* = 0.64, *r* = 0.80, Figure 7d), representing a 45% gain in *R2* and matching the performance of multi-state docking results based on the PDB 5VA2-derived model.

These findings suggest that the predictive power of computational drug docking is enhanced not merely by the accuracy of individual models, but by the structural diversity and complementarity provided by an ensemble of protein conformations. Rather than relying solely on a single experimentally determined protein structure, the ensemble benefits from incorporating AlphaFold-predicted models that capture alternative conformations identified through our state-specific sampling approach. These diverse protein models reflect different structural features, which together offer a more comprehensive representation of the ion channel’s binding landscape and enhance the predictive performance of computational drug docking. Overall, these results reinforce that multi-state modeling offers a more realistic and predictive framework for understanding drug – ion channel interactions than traditional single-state approaches, emphasizing the value of both individual model evaluation and their collective integration. We are grateful for the reviewer’s suggestion.

(3) Figures where multiple datapoints are compared across states generally lack assessment of the statistical significance of observed trends (e.g. Figure 3d).

We appreciate the reviewer’s comment on the statistical significance assessment in Figure 3d. To clarify, the comparisons shown in the subpanels are based on three selected representative models for each state, rather than a broader population sample (similarly for Figure 3b). In the closed-state predicted models, the strong convergence of the voltagesensing domain (VSD), with an all-atom RMSD of 0.36 Å between cluster 1 and 2 closed-state sampling models and 0.95 Å to the outlier cluster, indicates minimal structural variation. Those RMSD values shown in the manuscript text demonstrates good convergence and by themselves represent statistical significance assessment of those models. This trend extends to open-state and inactivated-state AlphaFold models with similarly limited differences in the VSD regions among them. This convergence suggests that population-based statistical analysis may not reveal meaningful deviations, as the low variability among models limits the insights beyond those obtained from comparing representative structures.

Nonetheless, we acknowledge this limitation. In future studies, we plan to explore alternative modeling approaches to introduce greater variability, enabling a more robust statistical evaluation of state-specific trends in the predictions.

(4) Figure 3 and Figures S1-S4 compare structural differences between states. However, these differences are inferred from the initial models. The collection of conformations generated via the MD runs allow for much more robust comparisons of structural differences.

We have explored these conformational state dynamics through MD simulations for the Open (5VA2-based), Inactivated (AlphaFold Cluster 2), and Closed-state models, as presented in Figures S7, S8, S10, S11. These figures provide detailed insights: Figure S7-S8 analyzes SF and pore conformation dynamics, including averaged pore radii with and without voltage and superimposed conformational ensembles; Figure S10 tracks cross-subunit distances between protein backbone carbonyl oxygens, revealing sequential SF dilation steps near residues F627 an G628; and Figure S11 illustrates this SF dilation process over time, highlighting residue F627 carbonyl flipping and SF expansion. We appreciate the opportunity to clarify our approach.

**Reviewer #2 (Recommendations for the authors):**
Major concerns:(1) Protein fragments are used to model the closed and inactivated states of hERG, but the choices of fragments are not well justified. For instance, in Figure 1a, helices from 8EP1 (deactivated voltage-sensing domain) and a helix+loop from 5VA2 (selectivity filter) are used. Why just the selectivity filter and not the cytosolic domain, for instance? Why not some parts of the helices attached to the selectivity filter, or the whole membrane inserted domain of 8EP1? Same for the inactivated conformation in Figure 1c: why the cytosolic domain only?

We thank the reviewer for their thoughtful questions regarding our choice of protein fragments for modeling the closed and inactivated states of hERG in Figures 1a and 1c, and we appreciate the opportunity to justify these selections more clearly. Our approach to template selection was guided by our experience that providing AlphaFold2 with larger templates often leads it to overly constrain predictions to the input structure, reducing its flexibility to explore alternative conformations. In contrast, smaller, targeted fragments increase the likelihood that AlphaFold2 will incorporate the desired structural features while predicting the rest of the protein. We have provided a more detailed discussion of this in the methodology section.

For the closed state (Figure 1a), we chose the deactivated voltage-sensing domain (VSD) from the rat EAG channel (PDB 8EP1) to inspire AlphaFold2 to predict a similarly deactivated VSD conformation characteristic of hERG channel closure, as this domain’s downward shift is a hallmark of potassium channel closure. We paired this with the selectivity filter (SF) and adjacent residues from the open-state hERG structure (PDB 5VA2) to maintain its conductive conformation, as it is generally understood that K^+^ channel closure primarily involves the intracellular gate rather than significant SF distortion. Including additional helices (e.g., S5–S6) or the entire membrane domain from PDB 8EP1 risked biasing the model toward the EAG channel’s pore structure, which differs from hERG’s, while omitting the cytosolic domain ensured focus on the VSD-driven closure without over-constraining cytoplasmic domain interactions.

For the inactivated state (Figure 1c), we initially used only the cytosolic domain from PDB 5VA2 to anchor the prediction while allowing AlphaFold2 to freely sample transmembrane domain conformations, particularly the SF, where the inactivation occurs via its distortion. Excluding the SF or attached helices at this stage avoided locking the model into the open-state SF, and the cytosolic domain alone provided a minimal scaffold to maintain hERG’s intracellular architecture without dictating pore dynamics. Following the initial prediction, we initiated more extensive sampling by using one of the predicted SFs that differs from the open-state SF (PDB 5VA2) as a structural seed, aiming to guide predictions away from the open-state configuration. The VSD and cytosolic domain were also included in this state to discourage pore closure during prediction. Using larger fragments, like the full membrane-spanning domains or additional cytosolic regions from the open-state structure might reduce AlphaFold2’s ability to deviate from the open-state conformation, undermining our goal of capturing more diverse, state-specific features.

It is worth noting that multiple strategies could potentially achieve the predicted models in our study, and here we only present examples of the paths we took and validated. It is likely that many of the steps may be unnecessary and could be skipped, and future work building on our approach can further explore and streamline this process. A consistent theme underlies our choices: for the closed state, we know the VSD should adopt a deactivated (“down”) conformation, so we provide AlphaFold2 with a specific fragment to guide this outcome; for the inactivated state, we recognize that the SF must change to a non-conductive conformation, so we grant AlphaFold2 flexibility to explore diverse conformations by minimizing initial constraints on the transmembrane region.

With greater sampling and computational resources, it is possible we could identify additional plausible, non-conductive conformations that might better represent an inactivated state, as hERG inactivation may encompass a spectrum of states. In this study, due to resource limitations, we focused on generating and validating a subset of conformations. Still, we acknowledge that broader exploration could further refine these models, which could be pursued in future studies. We updated the Methods and Discussion sections to reflect this perspective, and we are grateful for the reviewer’s input, which encourages us to clarify our rationale and highlight the adaptability of our approach.

To demonstrate the broader feasibility of this approach, we applied it to another ion channel system, voltage-gated sodium channel Na_V_ 1.5, as illustrated in Figure S14. In this example, a deactivated VSD II from the cryo-EM structure of a homologous ion channel Na_V_1.7 (PDB 6N4R) (DOI: 10.1016/j.cell.2018.12.018), which was trapped in a deactivated state by a bound toxin, was used as a structural template. This guided AlphaFold to generate a Na_V_1.5 model in which all four voltage sensor domains (VSD I–IV) exhibit S4 helices in varying degrees of deactivation. Compared to the cryo-EM openstate Na_V_1.5 structure (PDB 6LQA) (DOI: 10.1002/anie.202102196), the predicted model displays a visibly narrower pore, representing a plausible closed state. This example underscores the versatility of our strategy in modeling alternative conformational states across diverse ion channels.

(2) While the authors rely on AF2 (ColabFold) for the closed and inactivated states, they use Rosetta to model loops of the open state. Why not just supply 5VA2 as a template to ColabFold and rebuild the loops that way? Without clear explanations, these sorts of choices give the impression that the authors were looking for specific answers that they knew from their extensive knowledge of the hERG system. While the modeling done in this paper is very nice, its generalizability is not obvious.

We appreciate the reviewer’s question about our use of Rosetta to model loops in the open-state hERG channel (PDB5VA2) rather than rebuilding it entirely with ColabFold. In the study, we conducted a control experiment supplying parts of PDB 5VA2 to ColabFold to rebuild the loops, generating 100 models (Figure 2a: predicted open state). The top-ranked model (by pLDDT) differed from our Rosetta-modelled structure by only 0.5 Å RMSD, primarily due to the flexible extracellular loops as expected, with the pore and selectivity filter (our areas of focus) remaining nearly identical. We chose the Rosetta-refined cryo-EM structure as this structure and approach have been widely used as an open-state reference in our other hERG channel studies, such as by Miranda et al. (DOI: 10.1073/pnas.1909196117) and Yang et al. (DOI: 10.1161/CIRCRESAHA.119.316404), to ensure that our results are more directly comparable to prior work in the field. Nonetheless, as both models (with loops modeled by Rosetta or AlphaFold) were virtually identical, we would expect no significant differences if either were used to represent the open state in our study. We have incorporated this clarification into the main text.

(3) pLDDT scores were used as a measure of reliable and accurate predictions, but plDDT is not always reliable for selecting new/alternative conformations (see https://doi.org/10.1038/s41467-024-515072 and https://www.nature.com/articles/s41467-024-51801-z).

We acknowledge that while pLDDT is a valuable indicator of structural confidence in AlphaFold2 predictions, its limitations warrant consideration. In our revision, we mitigated this by not relying solely on pLDDT, but we also performed protein backbone dihedral angle analysis of the protein regions of focus in all predicted models to ensure comprehensive coverage of conformational variations. From our AlphaFold modeling results, we tested a model from cluster 3 of the inactivated-state sampling process, which exhibited lower pLDDT scores, and included these results in our revised analysis. We included a note in the revised manuscript’s Discussion section: “As noted in recent studies, pLDDT scores are not reliable indicators for selecting alternative conformations (DOI: 10.1038/s41467-024-51507-2 and DOI: 10.1038/s41467-024-51801-z). To address this, we performed a protein backbone dihedral angle analysis in the regions of interest to ensure that our evaluation captured a representative range of sampled conformations.”

(4) Extensive work has been done using AF2 to model alternative protein conformations (https://www.biorxiv.org/content/10.1101/2024.05.28.596195v1.abstract, along with some references the authors cite, such as work by McHaourab); another group recently modeled the ion channel GLIC (https://www.biorxiv.org/content/10.1101/2024.09.05.611464v1.abstract). Therefore, this work, though generally solid and thorough, seems more like a variation on a theme than a groundbreaking new methodology, especially because of the generalizability issues mentioned above.

We sincerely thank the reviewer for acknowledging the solidity of our study and for drawing our attention to the impressive recent efforts using AlphaFold2 to explore alternative protein conformations. These studies are valuable contributions that highlight the versatility of AlphaFold2, and we are grateful for their context in evaluating our work.

Building on these efforts, our approach not only enhances the prediction of conformational diversity but also introduces a twist by incorporating structural templates to guide AlphaFold2 toward specific functional protein states. More significantly, our study advances beyond mere structural modeling by integrating these conformations with their rigorous validation by incorporating multiple simulation results tested against experimental data to reveal that AlphaFold-predicted conformations can align with distinct physiological ion channel states. A key finding is that drug binding predictions using AlphaFold-derived hERG channel states substantially improve correlation with experimental data, which is a longstanding challenge in computational screening of multi-state proteins like the hERG channel, for which previous structural models have been mostly limited to the open state based on the cryo-EM structures. Our approach not only captures this critical state dependence but also reveals potential molecular determinants underlying enhanced drug binding during hERG channel inactivation, a phenomenon observed experimentally but poorly understood. These insights advance drug safety assessment by improving predictive screening for hERG-related cardiotoxicity, a major cause of drug attrition and withdrawal.

We view our methodology as a natural evolution of the advancements cited by the reviewer, offering an approach that predicts diverse hERG channel conformational states and links them to meaningful functional and pharmacological outcomes. To address the reviewer’s concern about generalizability, we have expanded the methodology section to make it easier to follow and include additional details. As an example, we show how our approach can be applied to model another ion channel system, Na_V_1.5, in Figure S14.

Furthermore, to enhance the applicability of our methodology, we have uploaded the scripts for analyzing AlphaFoldpredicted models to GitHub (https://github.com/k-ngo/AlphaFold_Analysis), ensuring they are adaptable for a wide range of scenarios with extensive documentation. This enables users, even those not focused on ion channels, to effectively apply our tools to analyze AlphaFold predictions for their own projects and produce publication-ready figures.

While it is likely that multiple modeling approaches could lead AlphaFold to model alternative protein conformations, the key challenge lies in validating the physiological relevance of those predicted states. This study is intended to support other researchers in applying our template-guided approach to different protein systems, and more importantly, in rigorously *in silico* testing and validation of the biological significance of the conformation-specific structural models they generate.

Minor concerns:(1) The authors mention in the Introduction section that capturing conformational states, especially for membrane proteins that may be significant as drug targets, is crucial. It would be helpful to relate their work to the NMR studies domains of the hERG channel, particularly the N-terminal “eag” domain, which is crucial for channel function and can provide insights into conformational changes associated with different channel states (https://doi.org/10.1016/j.bbrc.2010.10.132).

We appreciate the reviewer’s insightful comment regarding the PAS domain and the potential influence of other regions, such as the N-linker and distal C-region, on drug binding and state transitions.

The PAS domain did appear in the starting templates used for initial structural modeling (as shown in Figure 1a, b, c), but it was not included in the final models used for subsequent analyses. The omission was primarily due to hardwareimposed constraints, as including these additional regions would exceed the memory capacity of our current graphics processing unit (GPU) card, leading to failures during the prediction step.

The PAS domain, even if not serving as a conventional direct drug-binding site, can influence the gating kinetics of hERG channels. By altering the probability and duration with which channels occupy specific states, it can indirectly affect how well drugs bind. For example, if the presence of the PAS domain shifts hERG channel gating so that more channels enter (and remain in) the inactivated state as was shown previously (e.g., DOI: 10.1085/jgp.201210870), drugs with a higher affinity for that state would appear to bind more potently, as observed in previous electrophysiological experiments (e.g., DOI: 10.1111/j.1476-5381.2011.01378.x). It is also plausible that the PAS domain could exert allosteric effects that alter the conformational landscape of the hERG channel during gating transitions, potentially impacting drug accessibility or binding stability. This is an intriguing hypothesis and an important avenue for future research.

With access to more powerful computational resources, it would be valuable to explore the full-length hERG channel, including the PAS domain and associated regions, to assess their potential contributions to drug binding and gating dynamics. We incorporated a discussion of these points into the main text, acknowledging the limitations of our current models and highlighting the need for future studies to explore these regions in greater detail. The addition reads: “…Our models excluded the N-terminal PAS domain due to GPU memory limitations, despite its inclusion in initial templates. This omission may overlook its potential roles in gating kinetics and allosteric effects on drug binding (e.g., PMID: 21449979, PMID: 23319729, PMID: 29706893, PMID: 30826123, DOI:10.4103/jpp.JPP_158_17). Future research will explore the full-length hERG channel with enhanced computational resources to assess these regions’ contributions to conformational state transitions and pharmacology.”

(2) In the second-to-last paragraph of the Introduction, the authors describe how AlphaFold2 works. They write, “AlphaFold2 primarily requires the amino acid sequence of a protein as its input, but the method utilizes other key elements: in addition to the amino acid sequence, AlphaFold2 can also utilize multiple sequence alignments (MSAs) of similar sequences from different species, templates of related protein structures when available, and/or homologous proteins (Jumper et al., 2021a). Evolutionarily conserved regions over multiple isoforms and species indicated that the sequence is crucial for structural integrity”. The last sentence is confusing; if the authors mean that all information required to fold the protein into its 3D structure is present in its primary sequence, that has been the paradigm. It is unclear from this paragraph what the authors wanted to convey.

We apologize for any confusion caused by this phrasing. Our intent was not to restate the well-established paradigm that a protein’s primary sequence contains the information needed for its 3D structure, but rather to emphasize how

AlphaFold2 leverages evolutionary conservation, via multiple sequence alignments (MSAs), to infer structural constraints beyond what a single sequence alone might reveal. Specifically, we aimed to highlight that conserved regions across species and isoforms provide additional context that AlphaFold2 uses to enhance the accuracy of its predictions, complementing the use of templates and homologous structures as described in Jumper et al. (2021). To clarify this, we revised the sentence in the manuscript to read: “AlphaFold2 primarily requires a protein's amino acid sequence as input, but it also leverages other critical data sources. In addition to the sequence, it incorporates multiple sequence alignments (MSAs) of related proteins from different species, available structural templates, and information on homologous proteins. While the primary sequence encodes the 3D structure, AlphaFold2 harnesses evolutionary conservation from MSAs to reveal structural insights that extend beyond what a single sequence can provide.” We thank the reviewer for pointing out this ambiguity.

(3) In the Results section, the authors state that the predictions generated by their method are evaluated by standard accuracy metrics, please elaborate - what standard metrics were used to judge the predictions and why (some references would be a nice addition). Further, on Page 6, the sentence “There are fewer differences between the open- and closed-state models (Figure S2b, d)” is confusing, fewer differences than what? or there are a few differences between the two states/models? Please clarify.

The original sentence referring to “standard accuracy metrics” is somewhat misplaced, as our intent was not to apply any conventional “benchmarking” to judge the predictions, but rather to evaluate functional and structural relevance in a physiologically meaningful context. Specifically, we assessed drug binding affinities from molecular docking simulations (in Rosetta Energy Units, R.E.U.) against experimental drug potency data (e.g., IC_50_ values converted to free energies in kcal/mol, Figure 7), analyzed differences in interaction networks across states in relation to known mutations affecting hERG inactivation (Figure 4, Table 2), validated ion conduction properties through MD simulations with the applied voltage against expected state-dependent hERG channel behavior (Figure 5), and compared predicted structural models to available experimental cryo-EM structures (Figure 3). We clarified in the text that our assessment emphasized the physiological plausibility of the generated conformations, drawing on evidence from existing computational and experimental studies at each step of the analysis above.

As for the sentence on page 6, “There are fewer differences between the open- and closed-state models,” we apologize for the ambiguity; we meant that the hydrogen bond networks in the selectivity filter region exhibit fewer differences between the open and closed states compared to the more pronounced variations seen between the open and inactivated states. We revised this sentence to read: “The open- and closed-state models show fewer differences in their selectivity filter hydrogen bond networks compared to those between the open and inactivated states,” to enhance readability.

(4) In the Discussion, the authors reiterate that this methodology can be extended to sample multiple protein conformations, and their system of choice was hERG potassium channel. I think this methodology can be applied to a system when there is enough knowledge of static structures, and some information on dynamics (through simulations) and mutagenesis analysis available. A well-studied system can benefit from such a protocol to gauge other conformational states.

We agree that this approach is well-suited to systems with sufficient static structures, dynamic insights from simulations, and mutagenesis data, as seen with the hERG channel. We appreciate the reviewer’s implicit concern about generalizability to less-characterized systems and addressed this in the Discussion as a limitation, noting that the method’s effectiveness may depend on prior knowledge. Future studies can explore whether the advent of AlphaFold3 and other deep learning approaches can enhance its applicability to systems with more limited data. We have added this comment to the Discussion: “…A limitation of our methodology is its reliance on well-characterized systems with ample static structures, molecular dynamics simulation data, and mutagenesis insights, as demonstrated with the hERG channel, which may limit its applicability to less-studied proteins.”

(5) The Methods section must be broken down into steps to make it easier to follow for the reader (if they want to implement these steps for themselves on their system of choice).a. Is possible to share example scripts and code used to piece templates together for AF2. Also, since the AF3 code is now available, the authors may comment on how their protocol can be applicable there or have plans to implement their protocol using AF3 (which is designed to work better for binding small molecules). Please see https://github.com/google-deepmind/alphafold3 for the recently released code for AF3.

We appreciate the reviewer’s suggestion to improve the Methods section and their comments on scripts and AlphaFold3 (AF3). We revised the Methods to separate it into clear steps (e.g., template preparation, AF2 setup, clustering, and refinement) for better readability and reproducibility, and uploaded the sample scripts along with the instructions to GitHub (https://github.com/k-ngo/AlphaFold_Analysis).

Regarding AF3’s recent code release, we plan to explore the applicability of our methodology to AF3 in a follow-up study, leveraging its advanced features to refine conformational predictions and state-specific drug docking, and added a brief comment to the Discussion to reflect this future direction: “…Following the recent release of AlphaFold3’s source code, we plan to explore the applicability of our template-guided methodology in a follow-up study, leveraging AF3’s advanced diffusion-based architecture to enhance protein conformational state predictions and state-specific drug docking, particularly given its improved capabilities for modeling small molecule – protein interactions…”

b. The authors modified the hERG protein by removing a segment, the N-terminal PAS domain (residues M1 - R397) because of graphics card memory limitation. Would the removal of the PAS domain affect the structure and function of the channel protein? HERG and other members of the “eag K^+^ channel” family contain a PAS domain on their cytoplasmic N terminus. Removal of this domain alters a physiologically important gating transition in HERG, and the addition of the isolated domain to the cytoplasm of cells expressing truncated HERG reconstitutes wild-type gating. (see https://doi.org/10.1371/journal.pone.0059265). Please elaborate on this.

We thank the reviewer for raising an important point about the removal of the N-terminal PAS domain and for highlighting its physiological role in hERG channel gating transitions. In our study, unlike experimental settings where PAS removal alters gating, we believe this omission has minimal impact on our key analyses.

The drug docking procedure focuses on optimizing drug binding poses with minor protein structural refinement around the putative drug binding site, which in our case is the hERG channel pore region, where hERG-blocking drugs predominantly bind. The cytoplasmic PAS domain, located distally from this site, remains outside the protein structure refinement zone during drug docking simulations. However, one aspect we have not yet considered is the potential effect of drug modulation of the hERG channel gating and vice versa particularly given the PAS domain’s role in gating. This interplay could be significant but requires investigation beyond our current drug docking framework. We plan to explore this in future studies using alternative simulation methodologies, such as extended MD simulations or enhanced sampling techniques, to comprehensively capture these dynamic protein - ligand interactions.

Similarly, in our 1 μs long MD simulations assessing ion conductivity (Figure 4), the timescale is too short for PASmediated gating changes to propagate through the protein and meaningfully influence ion conduction and channel activation dynamics, which occurs on a millisecond time scale (see e.g., DOI: 10.3389/fphys.2018.00207). To fully address this limitation, we plan to explore the inclusion of the PAS domain in a follow-up study with enhanced computational resources, allowing us to investigate its structural and functional contributions more comprehensively.

(6) The first paragraph of the Methods reads as though AF2 has layers that recycle structures. We doubt that the authors meant it that way. Please update the language to clarify that recycling is an iterative process in which the pairwise representation, MSA, and predicted structures are passed (“recycled”) through the model multiple times to improve predictions.

We agree that the phrasing might suggest physical layers recycling structures, which was not our intent. Instead, we meant to describe AlphaFold2’s iterative refinement process, where intermediate outputs, such as the pairwise residue representations, multiple sequence alignments (MSAs), and predicted structures, are iteratively passed (or “recycled”) through the model to enhance prediction accuracy. To clarify this, we revised the relevant sentence to read: “A critical feature of AlphaFold2 is its iterative refinement, where pairwise residue representations, MSAs, and initial structural predictions are recycled through the model multiple times, improving accuracy with each iteration.”

**Reviewer #3 (Recommendations for the authors):**
The authors should integrate the very recently published CryoEM experimental data of hERG inhibition by several drugs (Miyashita et al., Structure, 2024; DOI: 10.1016/j.str.2024.08.021).

We thank the reviewer for the suggestion. Here, we compare drug binding in our open-states (PDB 5VA2-derived and an additional AlphaFold-predicted model from Cluster 3 of inactivated-state-sampling attempt named “AF ic3”) and inactivated-state models, using the cationic forms of astemizole and E-4031, with the corresponding experimental structures (Figure S13). Drug binding in the closed state is excluded as the pore architecture deviates too much from those in the cryo-EM structures. Experimental data (DOI: 10.1124/mol.108.049056) indicate that both astemizole and E4031 bind more potently to the inactivated state of the hERG channel.

Astemizole (Figure S13a):

- In the PDB 5VA2-derived open-state model, astemizole binds centrally within the pore cavity, adopting a bent conformation that allows both aromatic ends of the molecule to engage in π–π stacking with the side chains of Y652 from two opposing subunits. Hydrophobic contacts are observed with S649 and F656 residues.

- In the AF ic3 open-state model, the ligand is stabilized through multiple π–π stacking interactions with Y652 residues from three subunits, forming a tight aromatic cage around its triazine and benzimidazole rings. Hydrophobic interactions are observed with hERG residues T623, S624, Y652, F656, and S660.

- In the inactivated-state model, astemizole adopts a compact, horizontally oriented pose deeper in the channel pore, forming the most extensive interaction network among all the states. The ligand is tightly stabilized by multiple π–π stacking interactions with Y652 residues across three subunits, and forms hydrogen bonds with residues S624 and Y652. Additional hydrophobic contacts are observed with residues F557, L622, S649, and Y652.

- Consistent with our findings, electrophysiology study by Saxena et al. identified hERG residues F557 and Y652 as crucial for astemizole binding, as determined through mutagenesis (DOI: 10.1038/srep24182).

- In the cryo-EM structure (PDB 8ZYO) (DOI: 10.1016/j.str.2024.08.021), astemizole is stabilized by π–π stacking with Y652 residues. However, no hydrogen bonds are detected which may reflect limitations in cryo-EM resolution rather than true absence of contacts. Additional hydrophobic interacts are observed with L622 and G648 residues.

E-4031 (Figure S13b):

- In the PDB 5VA2-derived open-state model, E-4031 binds within the central cavity primarily through polar interactions. It forms a π–π stacking interaction with residue Y652, anchoring one end of the molecule. Polar interactions are observed with residues A653 and S660. Additional hydrophobic contacts are observed with residues A652 and Y652.

- In the AF ic3 open-state model, E-4031 adopts a slightly deeper pose within the central cavity stabilized by dual π–π stacking interactions between its aromatic rings and hERG residue Y652. Additional hydrogen bonds are observed with residues S624 and Y652, and hydrophobic contacts are observed with residues T623 and S624.

- In the inactivated-state model, E-4031 adopts its deepest and most stabilized binding pose, consistent with its experimentally observed preference for this state. The ligand is stabilized by multiple π–π stacking interactions between its aromatic rings and hERG residues Y652 from opposing subunits. The sulfonamide nitrogen engages in hydrogen bonding with residue S649, while the piperidine nitrogen hydrogen bonds with residue Y652. Hydrophobic contacts with residues S624, Y652, and F656 further reinforce the binding, enclosing the ligand in a densely packed aromatic and polar environment.

- Previous mutagenesis study showed that mutations involving hERG residues F557, T623, S624, Y652, and F656 affect E-4031 binding (DOI: 10.3390/ph16091204).

- In the cryo-EM structure (PDB 8ZYP) (DOI: 10.1016/j.str.2024.08.021), E-4031 engages in a single π–π stacking interaction with hERG residue Y652, anchoring one end of the molecule. The remainder of the ligand is stabilized predominantly through hydrophobic contacts involving residues S621, L622, T623, S624, M645, G648, S649, and additional Y652 side chains, forming a largely nonpolar environment around the binding pocket.

In both cryo-EM structures, astemizole and E-4031 adopt binding poses that closely resembles the inactivated-state model in our docking study, consistent with experimental evidence that these drugs preferentially bind to the inactivated state (DOI: 10.1124/mol.108.049056). This raises the possibility that the cryo-EM structures may capture an inactivatedlike channel state. However, closer examination of the SF reveals that the cryo-EM conformations more closely resemble the open-state PDB 5VA2 structure (DOI: 10.1016/j.cell.2017.03.048), which has been shown to be conductive here and in previous studies (DOI: 10.1073/pnas.1909196117, 10.1161/CIRCRESAHA.119.316404).

The conformational differences between the cryo-EM and open-state docking results may reflect limitations of the docking protocol itself, as GALigandDock assumes a rigid protein backbone and cannot account for ligand-induced large conformational changes. In our open-state models, the hydrophobic pocket beneath the selectivity filter is too small to accommodate bulky ligands (Figure 3a, b), whereas the cryo-EM structures show a slight outward shift in the S6 helix that expands this space (Figure S13).These allosteric rearrangements, though small, falls outside the scope of the current docking protocol, which lacks flexibility to capture these local, ligand-induced adjustments (DOI: 10.3389/fphar.2024.1411428).

In contrast, docking to the AlphaFold-predicted inactivated-state model reveals a reorganization beneath the selectivity filter that creates a larger cavity, allowing deeper ligand insertion. Notably, neither our inactivated-state docking nor the available cryo-EM structures show strong interactions with F656 residues. However, in the AlphaFold-predicted inactivated model, the more extensive protrusion of F656 into the central cavity may further occlude the drug’s egress pathway, potentially trapping the ligand more effectively. This could explain why mutation of F656 significantly reduces the binding affinity of E-4031 (DOI: 10.3390/ph16091204). These findings suggest that inactivation may trigger a series of modular structural rearrangements that influence drug access and binding affinity, with different aspects potentially captured in various computational and experimental studies, rather than resulting from a single, uniform conformational change.

Discussion of the original Wang and Mackinnon finding, DOI: 10.1016/j.cell.2017.03.048 regarding C-inactivation, pore mutation S631A and F627 rearrangement is likely warranted. Since hERG inactivation is present at 0 mV in WT channels (the likely voltage for the CryoEM study) please discuss how this might affect interpretations of starting with this structure as a template for models presented here, perhaps as part of Figure S1.

We sincerely thank the reviewer for bringing up the insightful findings from Wang and MacKinnon regarding hERG C-type inactivation as well as the voltage context of their cryo-EM structure (PDB 5VA2). We recognize that WT hERG exhibits inactivation at 0 mV, likely the condition of the cryo-EM study, raising the possibility that PDB 5VA2, while classified as an open state, might subtly reflect features of inactivation. Notably, PDB 5VA2 has been widely adopted in numerous studies and consistently found to represent a conducting state, such as in Yang et al. (DOI: 10.1161/CIRCRESAHA.119.316404) and Miranda et al. (DOI: 10.1073/pnas.1909196117). Our MD simulations further support this, showing K^+^ conduction in the 5VA2-based open-state model (Figure 4a, c), consistent with its selectivity filter conformation (Figure S1a). Although we used PDB 5VA2 as a starting template for predicting inactivated and closed states, our AlphaFold2 predictions did not rigidly adhere to this structure, as evidenced by distinct differences in hydrogen bond networks, drug binding affinities, pore radii, and ion conductivity between our state-specific hERG channel models (Figures S2, 5, 3b, 4). Nevertheless, this does not preclude the possibility that PDB 5VA2’s certain potential inactivated-like traits at 0 mV could subtly influence our predictions elsewhere in the model, which warrants further exploration in future studies. In our revised analysis, we also tested an alternative AlphaFold-predicted conformation, referred to as Open (AlphaFold cluster 3), which, while sharing some similarities with PDB 5VA2, exhibits subtle differences in the selectivity filter and pore conformations. This structure was also found to be conducting ions and showed a drug binding profile similar to that of the PDB 5VA2-based open-state model. We greatly appreciate this feedback which helped us refine and strengthen our analysis.

Page 8, the significance of 750 and 500 mV in terms of physiological role?

We appreciate this opportunity to clarify the methodological rationale. Although these voltages significantly exceed typical physiological membrane potentials, their use in MD simulations is a well-established practice to accelerate ion conduction events. This approach helps overcome the inherent timescale limitations of conventional MD simulations, as demonstrated in previous studies of hERG and other ion channels. For instance, Miranda et al. (DOI: 10.1073/pnas.1909196117), Lau et al. (DOI: 10.1038/s41467-024-51208-w), Yang et al. (DOI: 10.1161/CIRCRESAHA.119.316404) applied similarly high voltages (500~750 mV) to study hERG K^+^ conduction, which is notably small under physiological conditions at ~2 pS (DOI: 10.1161/01.CIR.94.10.2572), necessitating amplification to observe meaningful permeation within nanosecond-to-microsecond timescales. Likewise, studies of other K^+^ ion channels, such as Woltz et al. (DOI: 10.1073/pnas.2318900121) on small-conductance calcium-activated K^+^ channel SK2 and Wood et al. (DOI: 10.1021/acs.jpcb.6b12639) on Shaker K^+^ channel, have used elevated voltages (250~750 mV) to probe ion conduction mechanisms via MD simulations. In addition, the typical timescale of these simulations (1 μs) is too short to capture major structural effects such as those leading to inactivation or deactivation which occur over milliseconds in physiological conditions.

The abstract could be edited a bit to more clearly state the novel findings in this study.

We thank the reviewer for their suggestion. We have revised the abstract to read: “To design safe, selective, and effective new therapies, there must be a deep understanding of the structure and function of the drug target. One of the most difficult problems to solve has been resolution of discrete conformational states of transmembrane ion channel proteins. An example is K_V_11.1 (hERG), comprising the primary cardiac repolarizing current, *Ikr*. hERG is a notorious drug antitarget against which all promising drugs are screened to determine potential for arrhythmia. Drug interactions with the hERG inactivated state are linked to elevated arrhythmia risk, and drugs may become trapped during channel closure. While prior studies have applied AlphaFold to predict alternative protein conformations, we show that the inclusion of carefully chosen structural templates can guide these predictions toward distinct functional states. This targeted modeling approach is validated through comparisons with experimental data, including proposed state-dependent structural features, drug interactions from molecular docking, and ion conduction properties from molecular dynamics simulations. Remarkably, AlphaFold not only predicts inactivation mechanisms of the hERG channel that prevent ion conduction but also uncovers novel molecular features explaining enhanced drug binding observed during inactivation, offering a deeper understanding of hERG channel function and pharmacology. Furthermore, leveraging AlphaFold-derived states enhances computational screening by significantly improving agreement with experimental drug affinities, an important advance for hERG as a key drug safety target where traditional single-state models miss critical state-dependent effects. By mapping protein residue interaction networks across closed, open, and inactivated states, we identified critical residues driving state transitions validated by prior mutagenesis studies. This innovative methodology sets a new benchmark for integrating deep learning-based protein structure prediction with experimental validation. It also offers a broadly applicable approach using AlphaFold to predict discrete protein conformations, reconcile disparate data, and uncover novel structure-function relationships, ultimately advancing drug safety screening and enabling the design of safer therapeutics.”

Many of the Supplemental figures would fit in better in the main text, if possible, in my opinion. For instance, the network analysis (Fig. S2) appears to be novel and is mentioned in the abstract so may fit better in the main text. The discussion section could be focused a bit more, perhaps with headers to highlight the key points.

Yes, we agree with the reviewer and made the suggested changes. We moved Figure S2 as a new main-text figure.

Additionally, we revised the Discussion section to improve focus and clarity.